# Faster Algorithms for Structured John Ellipsoid Computation

**Yang Cao**
Wyoming Seminary
ycao4@wyomingseminary.org

**Xiaoyu Li**
University of New South Wales
7.xiaoyu.li@gmail.com

**Zhao Song**
University of California, Berkeley
magic.linuxkde@gmail.com

**Xin Yang**
The University of Washington
yangxin199207@gmail.com

**Tianyi Zhou**
University of Southern California
tzhou029@usc.edu

## Abstract

The famous theorem of Fritz John states that any convex body has a unique maximal volume inscribed ellipsoid, known as the John Ellipsoid. Computing the John Ellipsoid is a fundamental problem in convex optimization. In this paper, we focus on approximating the John Ellipsoid inscribed in a convex and centrally symmetric polytope defined by $P := \{x \in \mathbb{R}^d : -\mathbf{1}_n \leq Ax \leq \mathbf{1}_n\}$, where $A \in \mathbb{R}^{n \times d}$ is a rank-$d$ matrix and $\mathbf{1}_n \in \mathbb{R}^n$ is the all-ones vector. We develop two efficient algorithms for approximating the John Ellipsoid. The first is a sketching-based algorithm that runs in nearly input-sparsity time $\widetilde{O}(\mathrm{nnz}(A) + d^\omega)$, where $\mathrm{nnz}(A)$ denotes the number of nonzero entries in the matrix $A$ and $\omega \approx 2.37$ is the current matrix multiplication exponent. The second is a treewidth-based algorithm that runs in time $\widetilde{O}(n\tau^2)$, where $\tau$ is the treewidth of the dual graph of the matrix $A$. Our algorithms significantly improve upon the state-of-the-art running time of $\widetilde{O}(nd^2)$ achieved by [Cohen, Cousins, Lee, and Yang, COLT 2019].

## 1 Introduction

The concept of the John Ellipsoid, introduced in the seminal work of [Joh48], plays a fundamental role in convex optimization and convex geometry [Bal91, Bal01, LYZ05, Tod16]. John's theorem states that every compact convex set with a nonempty interior has a unique maximum-volume inscribed ellipsoid, known as the John Ellipsoid [Joh48]. The John Ellipsoid has numerous significant applications, including high-dimensional sampling [Vem05, CDWY18, GN23], linear programming [LS14], online learning [BCBK12, HK16], differential privacy [NTZ13], and uncertainty quantification [TLY24]. Moreover, it is known that computing the John Ellipsoid is equivalent to the D-optimal design problem in statistics [Puk06, Tod16], which has a lot of applications in machine learning [AZLSW17, WYS17, LFN18].

In this paper, we study the problem of computing the John ellipsoid $Q$ of a convex and centrally symmetric polytope $P := \{x \in \mathbb{R}^d : -\mathbf{1}_n \leq Ax \leq \mathbf{1}_n\}$, where $A \in \mathbb{R}^{n \times d}$ is a rank-$d$ matrix and $\mathbf{1}_n$ is the all-ones vector. The John Ellipsoid $E$ is the unique solution to the optimization problem $\max_{Q \subseteq \mathcal{E}^d} \mathrm{vol}(Q)$ s.t. $Q \subseteq P$, where $\mathcal{E}^d$ is the set of all ellipsoids in $\mathbb{R}^d$ and $\mathrm{vol}(Q)$ denotes the volume of $Q$. Since this geometric optimization problem can be formulated as a constrained

convex optimization problem, the John Ellipsoid can be computed in polynomial time using convex optimization solvers, such as first-order methods [Kha96, KY05] and second-order interior-point methods [NN94, SF04]. The most efficient algorithm using convex optimization solvers takes $O(nd^3)$ time, as demonstrated by [KY05, TY07].

Recently, [CCLY19] proposed a simple and fast fixed-point iteration (Algorithm 1) to compute the John Ellipsoid in $\widetilde{O}(nd^2)$ time by reducing the problem to the computation of $\ell_\infty$ Lewis weights of the matrix $A$. The $\ell_\infty$ Lewis weights of $A$ is a vector $w \in \mathbb{R}^n$ which can be seen as a weighted version of the leverage scores of $A$.

---

**Algorithm 1** Approximating John Ellipsoid inside symmetric polytopes, Algorithm 1 [CCLY19]

---

1: **procedure** APPROXJE($A \in \mathbb{R}^{n \times d}$)
2:      $w_1 \leftarrow (d/n) \cdot \mathbf{1}_n$
3:      **for** $k = 1, \cdots, T-1$ **do**
4:          **for** $i = 1 \rightarrow n$ **do**
5:              $w_{k+1,i} = w_{k,i} \cdot a_i^\top (A^\top \operatorname{diag}(w_k) A)^{-1} a_i$
6:          **end for**
7:      **end for**
8:      **for** $i = 1 \rightarrow n$ **do**
9:          $v_i = \frac{1}{T} \sum_{k=1}^{T} w_{k,i}$
10:      **end for**
11:      $U \leftarrow \operatorname{diag}(u)$
12:      **return** $A^\top U A$
13: **end procedure**

---

This iterative approach plays a crucial role in simplifying the computation of the John Ellipsoid for convex symmetric polytopes defined by a set of inequalities. Delving deeper into the algorithmic intricacies of [CCLY19], it becomes evident that a primary computational hurdle lies in calculating the quadratic forms, denoted as $a^\top B^{-1} a$, where $B$ is a weighted version of $A^\top A$ and $a$ is a row vector in $A$. The algorithm in [CCLY19] employs the standard linear algebraic approach, which involves computing the Cholesky decomposition and subsequently solving linear systems. However, a significant drawback of this method is its time complexity $\widetilde{O}(nd^2)$. In many computational scenarios, this can be excessively time-consuming.

To address this challenge, we develop a new sketching-based algorithm that offers an improved running time compared to [CCLY19]. Furthermore, we also provide an algorithm that exploit certain special structures to achieve speedups.

## 1.1 Algorithm in Nearly Input-Sparsity Time

Our first contribution is an algorithm that computes the John Ellipsoid in nearly input-sparsity time.

**Theorem 1.1** (Main result I, input-sparsity time). *Given a matrix $A \in \mathbb{R}^{n \times d}$, let a symmetric convex polytope be defined as $P := \{x \in \mathbb{R}^d : -\mathbf{1}_n \leq Ax \leq \mathbf{1}_n\}$. For any $\epsilon, \delta \in (0, 0.1)$, where $\delta$ denotes the failure probability, there exists a randomized algorithm (Algorithm 2) that with probability at least $1 - \delta$ outputs an ellipsoid $Q$ satisfying $\frac{1}{\sqrt{1+\epsilon}} \cdot Q \subseteq P \subseteq \sqrt{d} \cdot Q$. Moreover, it runs within $O(\epsilon^{-1} \log(n/d))$ iterations and each iteration takes $\widetilde{O}(\epsilon^{-1} \operatorname{nnz}(A) + \epsilon^{-2} d^\omega)$ time, where $\operatorname{nnz}(A)$ is the number of non-zero entries of $A$ and $\omega \approx 2.37$ denotes the current matrix multiplication exponent [ADW+24], and the $\widetilde{O}$ hides the $\log(d/\delta)$ factor.*

Compared to [CCLY19], we have significantly improved the per-iteration cost, reducing it from $O(nd^2)$ to $\widetilde{O}(\epsilon^{-1}\operatorname{nnz}(A) + \epsilon^{-2} d^\omega)$. Here, the $\widetilde{O}$-notation hides the $\log(d/\delta)$ factor. When the matrix $A$ is sparse, our algorithm significantly outperforms [CCLY19]. Note that when the matrix $A$ is dense, i.e., $\operatorname{nnz}(A) = \Theta(nd)$, our per-iteration cost becomes $\widetilde{O}(\epsilon^{-1} nd + \epsilon^{-2} d^\omega)$. In the regime where $n > d^\omega$ and $d > \epsilon^{-1}$, our algorithm is also better than [CCLY19] even when the matrix $A$ is dense.

The technical improvements stem from two key factors: First, to achieve an input-sparsity running time, we introduce an additional subsampling procedure alongside the sketching approach used by [CCLY19]. This sampling step utilizes an approximation of leverage scores, significantly accelerating

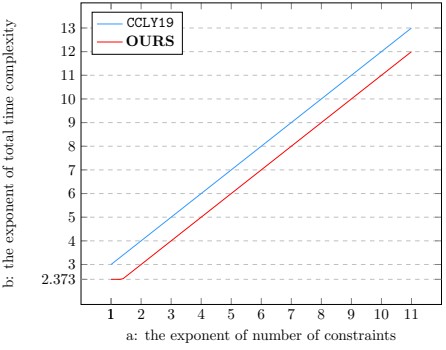

Figure 1: Time complexity comparison between CCLY19 (denotes [CCLY19]) and ours, assuming $n = d^a$, $\epsilon = \Theta(1)$, and ignoring the log factors. The $x$-axis is corresponding to $a$ and $y$-axis is corresponding to $b$. The $n^b$ is the total running time.

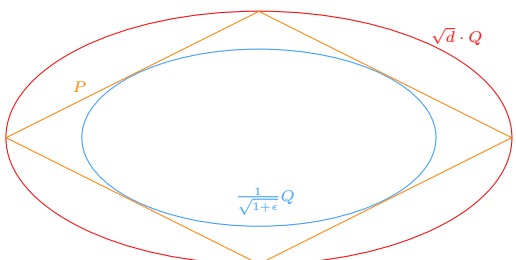

Figure 2: The geometric interpretation of the output ellipsoid. Let $P$ be a given input polytope. We can find an ellipsoid $Q$ so that $\frac{1}{\sqrt{1+\epsilon}}Q \subseteq P \subseteq \sqrt{d} \cdot Q$.

the computation of matrix inverses, which is the main bottleneck in the per-iteration complexity of [CCLY19]. Second, our approach requires a more detailed analysis to manage the accumulation of errors from both sampling and sketching within each iteration, thereby reducing the explicit dimension dependency in running time from $n$ to $\log(n)$.

## 1.2 Algorithm for Small Treewidth

Our second algorithm is a treewidth-based algorithm to compute the John Ellipsoid, which is extremely faster when the matrix $A$ has small treewidth. Informally speaking, treewidth is a property that measures how "tree-like" a graph is, and it originates from the structural graph theory [BGHK95, Dav06, LMS13]. For a matrix $A$, the concept of treewidth is associated with the dual graph $G_A$ that is constructed from the matrix $A$. We defer the formal definitions to Section 2.3. We state our second main result as follows.

**Theorem 1.2** (Main result II, small treewidth). *Given a matrix $A \in \mathbb{R}^{n \times d}$ whose dual graph $G_A$ has treewidth $\tau$, let a symmetric convex polytope be defined as $P := \{x \in \mathbb{R}^d : -\mathbf{1}_n \leq Ax \leq \mathbf{1}_n\}$. For any $\epsilon, \delta \in (0, 0.1)$, where $\delta$ denotes the failure probability, there exists a deterministic algorithm (Algorithm 3) that outputs an ellipsoid $Q$ satisfying*

$$\frac{1}{\sqrt{1+\epsilon}} \cdot Q \subseteq P \subseteq \sqrt{d} \cdot Q.$$

*Moreover, it runs within $O(\epsilon^{-1} \log(n/d))$ iterations and each iteration takes $O(n\tau^2)$ time.*

Our treewidth-based algorithm is extremely useful when the input matrix $A$ has small treewidth. In many real world datasets, the input matrix $A$ typically can have large dimension on $n$ and $d$, but it often exhibits small treewidth. For example, in the Netlib dataset, most LP instances have sublinear treewidth, typically in the range $[d^{1/2}, d^{3/4}]$ [BDGR95]. In MATPOWER dataset used for power system analysis, the maximum problem size is $n = 20467, d = 12659$ while the maximum treewidth $\tau = 35$ [ZMST10, ZL21]. For a detailed experimental analysis of treewidth in real-world datasets, we refer the reader to [MSJ19].

It is also worth noting that having a small treewidth is a stricter condition compared to input sparsity since it places additional restrictions on the connectivity pattern of the matrix, which may not be captured solely by input sparsity.

**Roadmap.** The rest of the paper is organized as follows. In Section 2, we provide some preliminaries for treewidth and John Ellipsoid. In Section 3, we give the formal definition for the John Ellipsoid. In Section 4, we present the technique overview for this paper. In Section 5, we present our main algorithm (Algorithm 2) for approximating John Ellipsoid inside symmetric polytopes and show the running time for the algorithm. In addition, we prove the correctness of our implementation. In Section 6, we present our algorithm (Algorithm 3) for small treewidth setting. In Section 7, we provide the conclusion for our paper.

Table 1: Comparison of our algorithms with the previous state-of-the-art presented in [CCLY19]. Given the input matrix $A \in \mathbb{R}^{n \times d}$ and the approximation error $\epsilon \in (0, 0.1)$, our algorithms achieve less per-iteration cost while maintaining the same number of iterations. We ignore the $\widetilde{O}$-notation in the table.

| References | #Iters. | Cost per iter. |
|---|---|---|
| [CCLY19] | $\epsilon^{-1} \log(n/d)$ | $nd^2$ |
| Theorem 1.1 | $\epsilon^{-1} \log(n/d)$ | $\epsilon^{-1} \mathrm{nnz}(A) + \epsilon^{-2} d^\omega$ |
| Theorem 1.2 | $\epsilon^{-1} \log(n/d)$ | $n\tau^2$ |

## 2 Preliminaries

We first define some notations in Section 2.1. Then we introduce the definition of leverage score and its useful properties in Section 2.2. Next, we provides the necessary backgrounds of treewidth in Section 2.3. Then, in Section 2.4, we give the definition for Cholesky factorization. Finally, we state a matrix concentration bound in Section 2.5.

### 2.1 Notations

We use $\mathcal{N}(\mu, \sigma^2)$ to denote the normal distribution with mean $\mu$ and variance $\sigma^2$. Given two vectors $x$ and $y \in \mathbb{R}^d$, we use $\langle x, y \rangle$ to denote the inner product between $x$ and $y$, i.e., $\langle x, y \rangle = \sum_{i=1}^d x_i y_i$. We use $\mathbf{1}_n$ to denote an all-1 vector with dimension $n$. For any matrix $A \in \mathbb{R}^{d \times d}$, we say $A \succeq 0$ (positive semi-definite) if for all $x \in \mathbb{R}^d$ we have $x^\top A x \geq 0$. For a function $f$, we use $\widetilde{O}(f)$ to denote $f \cdot \mathrm{poly}(\log f)$. For a matrix $A$, we use $A^\top$ to denote the transpose of matrix $A$. We use $\omega \approx 2.371$ to denote the current matrix mulitpilcation exponent [ADW$^+$24]. For a matrix $A$, we use $\mathrm{nnz}(A)$ to denote the number of non-zero entries in $A$. For a square and full rank matrix $A$, we use $A^{-1}$ to denote the inverse of matrix $A$. For a positive integer, we use $[n]$ to denote the set $\{1, 2, \cdots, n\}$. For a vector $x$, we use $\|x\|_2$ to denote the entry-wise $\ell_2$ norm of $x$, i.e., $\|x\|_2 := (\sum_{i=1}^n x_i^2)^{1/2}$. We say a vector is $\tau$-sparse if it has at most $\tau$ non-zero entries. For a random variable $X$, we use $\mathbb{E}[X]$ to denote its expectation. We use $\Pr[\cdot]$ to denote the probability.

### 2.2 Leverage Score

We assume $A \in \mathbb{R}^{n \times d}$ has rank $d$. The leverage scores can be defined in several equivalent ways as follows.

**Definition 2.1** (Leverage score). *Given a matrix $A \in \mathbb{R}^{n \times d}$, let $U \in \mathbb{R}^{n \times d}$ be an orthonormal basis for the column space of $A$. For any $i \in [n]$, the leverage score of the $i$-th row of $A$ can be defined equivalently as: Part 1. $\sigma_i(A) = \|u_i\|_2$. Part 2. $\sigma_i(A) = a_i^\top (A^\top A)^{-1} a_i$. Part 3. $\sigma_i(A) = \max_{x \in \mathbb{R}^d} (a_i^\top x)^2 / \|Ax\|_2^2$.*

The last definition offers an intuitive understanding of leverage scores. A row $a_i$ has a higher leverage score when it is more influential, meaning there exists a vector $\mathbf{x}$ for which the inner product with $a_i$ is significantly larger than its average inner product (i.e., $\|A\mathbf{x}\|_2^2$) with the other rows of the matrix. This concept forms the basis of leverage score sampling, a widely used technique in which rows with higher leverage scores are sampled with greater probability.

Next, we state a well-known folklore property of leverage scores (see [SS11, CCLY19] for example).

**Lemma 2.2** (Folklore). *Given a matrix $A \in R^{n \times d}$, for any $i \in [n]$, it holds that $0 \leq \sigma_i(A) \leq 1$. Moreover, we have $\sum_{i=1}^n \sigma_i(A) = d$.*

We state a useful tool for leverage score from [DSW22], which proved a stronger version that computes the leverage score for the matrix in the form of $A(I - V^\top V)$. We only compute the leverage score for matrix $A$ here.

**Lemma 2.3** (Leverage score computation, Lemma 4.3 in [DSW22]). *Given a matrix $A \in \mathbb{R}^{n \times d}$, we can compute a vector $\widetilde{\sigma} \in \mathbb{R}^n$ in $\widetilde{O}(\epsilon_\sigma^{-2}(\mathrm{nnz}(A) + d^\omega))$ time, so that, $\widetilde{\sigma}$ is an approximation of the leverage score of matrix $A$, i.e., $\widetilde{\sigma} \in (1 \pm \epsilon_\sigma) \cdot \sigma(A)$, with probability at least $1 - \delta_\sigma$. The $\widetilde{O}$ hides the $\log(d/\delta_\sigma)$ factor.*

## 2.3 Treewidth

We first define the tree decomposition and treewidth of a given graph, see figure 3 for a concrete example.

**Definition 2.4** (Tree decomposition and tree width of a graph [BGHK95, Dav06, LMS13]). *A tree decomposition is a mapping of graphs into trees. For graph $G$, the tree decomposition is defined as pair $(M, T)$, where $T$ is a tree, and $M : V(T) \to 2^{V(G)}$ is a family of subsets of $V(G)$ called bags labelling the vertices of $T$, satisfies that:*

- *The vertices maintained by all bags is the same as those of graph G: $\cup_{t \in V(T)} M(t) = V(G)$.*

- *For every vertex $v \in V(G)$, the nodes $t \in V(T)$ satisfying $v \in M(t)$ induce a connected subgraph of $T$.*

- *For every edge $e = (u, v) \in E(G)$, there exist a node $t \in V(T)$ so that $u, v \in M(t)$.*

*where $V(\cdot)$ denotes the vertex set of a graph.*

*The width of a tree decomposition $(M, T)$ is $\max\{|M(t)| - 1 : t \in T\}$. The treewidth $\tau$ of $G$ is the minimum width over all tree decompositions of $G$.*

Given a matrix $A$, we generalize the definition of treewidth as the treewidth of its associated dual graph. Though the treewidth of a graph is NP-hard to compute [FLS⁺18, ACP87], it is possible to find a width-$O(\tau \log^3 n)$ tree decomposition within $O(m \operatorname{poly} \log n)$, where $m$ denotes the number of edges, $n$ denotes the number of vertices and $\tau$ denotes the treewidth of graph $G$ [BGS21].

**Definition 2.5** (Dual graph). *Given a matrix $A \in \mathbb{R}^{n \times d}$, we can optionally partition its rows into $m$ blocks of sizes $n_1, \ldots, n_m$ where $n = \sum_{i=1}^{m} n_i$. When no explicit block structure is given, we simply treat each row as its own block (i.e., $m = n$ and $n_i = 1$ for all $i$). The dual graph $G_A$ of the matrix $A$ is the graph $G_A = (V, E)$ with vertex set $V = \{1, \cdots, d\}$ (corresponding to the columns of $A$). We say an edge $(i, j) \in E$ if and only if there exists some row block $r \in [m]$ such that both $A_{r,i} \neq 0$ and $A_{r,j} \neq 0$, where $A_{r,i}$ denotes the submatrix of $A$ containing column $i$ and all rows in block $r$. The treewidth of the matrix $A$ is defined as the treewidth of its dual graph $G_A$.*

## 2.4 Cholesky Factorization

Next, we give the definition for Cholesky factorization.

**Definition 2.6** (Cholesky factorization). *Given a positive-definite matrix $P$, there exists a unique Cholesky factorization $P = LL^\top \in \mathbb{R}^{d \times d}$, where $L \in \mathbb{R}^{d \times d}$ is a lower-triangular matrix with real and positive diagonal entries.*

We then introduce a result based on the Cholesky factorization of a given matrix with treewidth $\tau$:

**Lemma 2.7** (Fast Cholesky factorization [BGHK95, Dav06]). *For any positive diagonal matrix $H \in \mathbb{R}^{n \times n}$, for any matrix $A^\top \in \mathbb{R}^{d \times n}$ with treewidth $\tau$, we can compute the Cholesky factorization $A^\top H A = LL^\top \in \mathbb{R}^{d \times d}$ in $O(n\tau^2)$ time, where $L \in \mathbb{R}^{d \times d}$ is a lower-triangular matrix with real and positive entries. $L$ satisfies the property that every row is $\tau$-sparse.*

**Remark 2.8.** *When only an $O(\log^3 n)$-approximation $\widetilde{\tau}$ to the treewidth $\tau$ is known (which can be computed in $O(m \operatorname{poly} \log n)$ time [BGS21]), the runtime becomes $O(n\widetilde{\tau}^2) = O(n\tau^2 \log^6 n)$, which remains efficient for small $\tau$.*

## 2.5 Matrix Concentration

We need the following matrix concentration bound as a tool to analyze the performance of our algorithm.

**Lemma 2.9** (Matrix Chernoff Bound [Tro11]). *Let $X_1, \ldots, X_s$ be i.i.d. symmetric random matrices with $\mathbb{E}[X_1] = 0$, $\|X_1\| \leq \gamma$ almost surely and $\|\mathbb{E}[X_1^\top X_1]\| \leq \sigma^2$. Let $C = \frac{1}{s} \sum_{i \in [s]} X_i$. For any $\epsilon \in (0, 1)$, it holds that $\Pr[\|C\| \geq \epsilon] \leq 2d \cdot \exp\left(-\frac{s\epsilon^2}{\sigma^2 + \gamma\epsilon/3}\right)$.*

# 3 Problem Formulation

In this section, we give the formal definition for the John Ellipsoid of a symmetric polytope. We first give a characterization of any symmetric polytope.

**Definition 3.1** (Symmetric convex polytope). *We define a symmetric convex polytope as*

$$P := \{x \in \mathbb{R}^d : |\langle a_i, x \rangle| \leq 1, \ \forall i \in [n]\}.$$

We define matrix $A \in \mathbb{R}^{n \times d}$ associated with the above polytope $P \subset \mathbb{R}^d$ as a collection of column vectors, i.e., $A = (a_1, a_2, \cdots, a_n)^\top$, and we assume $A$ is full rank. Note that since $P$ is symmetric, the John Ellipsoid of it must be centered at the origin. Since any origin-centered ellipsoid is of the form $\{x : x^\top G^{-2} x \leq 1\}$ for a positive definite matrix $G$, we can search over the optimal ellipsoid by searching over the possible matrix $G$. Note that for such an ellipsoid, the volume is proportional to $\det(G^{-1})^{1/2} = \det(G)^{-1/2}$, so maximizing the volume is equivalent to maximizing $\log(\det(G))^2 = 2\log(\det(G))$:

$$\text{Maximize} \log(\det(G))^2, \quad \text{subject to: } G \succeq 0 \quad \|Ga_i\|_2 \leq 1, \forall i \in [n] \tag{1}$$

In [CCLY19], it is shown that the optimal $G$ must satisfy $G^{-2} = A^\top \operatorname{diag}(w) A$, for the matrix $A$ and vector $w \in \mathbb{R}^n_{\geq 0}$. Thus, optimizing over $w$, we have the following optimization program:

$$\text{Minimize} \sum_{i=1}^n w_i - \log\det(\sum_{i=1}^n w_i a_i a_i^\top) - d, \quad \text{subject to: } w_i \geq 0, \ \forall i \in [n]. \tag{2}$$

For any weight vector $w \in \mathbb{R}^n_{\geq 0}$, we define the associated matrix

$$Q := \sum_{i=1}^n w_i a_i a_i^\top \in \mathbb{R}^{d \times d}. \tag{3}$$

Additionally, the optimality condition for this $w$ has been studied in [Tod16]:

**Lemma 3.2** (Optimality criteria, Proposition 2.5 in [Tod16]). *A weight $w \in \mathbb{R}^n$ is optimal for program (Eq. (2)) if and only if*

$$\sum_{i=1}^n w_i = d, \quad a_j^\top Q^{-1} a_j = 1, \text{ if } w_j \neq 0 \quad a_j^\top Q^{-1} a_j < 1, \text{ if } w_j = 0.$$

*where $Q$ is defined as in Eq. (3).*

Besides finding the exact John Ellipsoid, we can also find an $(1 + \epsilon)$-approximate John Ellipsoid:

**Definition 3.3** ($(1 + \epsilon)$-approximate John Ellipsoid). *For $\epsilon > 0$, we say $w \in \mathbb{R}^n_{\geq 0}$ is a $(1 + \epsilon)$-approximation of program (Eq. (2)) if $w$ satisfies $\sum_{i=1}^n w_i = d, \quad a_j^\top Q^{-1} a_j \leq 1 + \epsilon, \quad \forall j \in [n]$ where $Q$ is defined as in Eq. (3).*

Lemma 3.4 gives a geometric interpretation of the approximation factor in Definition 3.3. Note that for the exact John Ellipsoid $Q^*$ of the same polytope, $Q^* \subseteq P \subseteq \sqrt{d} \cdot Q^*$.

**Lemma 3.4** ($(1 + \epsilon)$-approximation is good rounding, Lemma 2.3 in [CCLY19]). *Let $P$ be defined as Definition 3.1. Let $w \in \mathbb{R}^n$ be a $(1 + \epsilon)$-approximation of Eq. (2), and let $Q$ be the associated matrix defined in Eq. (3). We define the ellipsoid $\mathcal{E} := \{x \in \mathbb{R}^d : x^\top Q x \leq 1\}$. Then the following property holds: $\frac{1}{\sqrt{1+\epsilon}} \cdot \mathcal{E} \subseteq P \subseteq \sqrt{d} \cdot \mathcal{E}$. Moreover, $\operatorname{vol}(\frac{1}{\sqrt{1+\epsilon}}\mathcal{E}) \geq \exp(-d\epsilon/2) \cdot \operatorname{vol}(\mathcal{E}^*)$ where $\mathcal{E}^*$ is the exact John Ellipsoid of $P$.*

# 4 Technical Overview

In Section 4.1, we provide a comprehensive overview of the framework from [CCLY19] upon which our work builds. In Section 4.2, we present our techniques for achieving nearly input-sparsity runtime. In Section 4.3, we describe our algorithm tailored for small treewidth.

## 4.1 Overview of Previous Work

The algorithm from [CCLY19] solves the John Ellipsoid problem via a fixed-point iteration scheme. Given the optimization program in Eq. (2), the optimal weight vector $w^*$ satisfies the fixed-point condition: for all $i \in [n]$, $w_{k+1,i} = w_{k,i} \cdot \sigma_i(w_k)$, where $\sigma_i(w) := a_i^\top (A^\top \operatorname{diag}(w) A)^{-1} a_i$. Starting from an initial weight $w_1 = (d/n) \cdot \mathbf{1}_n$, the algorithm iteratively updates the weights for $T = O(\epsilon^{-1} \log(n/d))$ iterations until convergence to an $(1 + \epsilon)$-approximate solution.

The main computational bottleneck in the naive fixed-point iteration is computing $\sigma_i(w_k)$ for all $i \in [n]$ at each iteration, which requires $O(nd^2)$ time using standard matrix inversion. To accelerate this, [CCLY19] applies a random Gaussian sketching matrix $S \in \mathbb{R}^{s \times d}$ (with $s = O(\epsilon^{-1})$) to approximate the quadratic form:

$$\sigma_i(w_k) = \|(A^\top \operatorname{diag}(w_k)A)^{-1/2} \sqrt{w_{k,i}} a_i\|_2^2 \approx \|S(A^\top \operatorname{diag}(w_k)A)^{-1/2} \sqrt{w_{k,i}} a_i\|_2^2.$$

Despite using sketching, [CCLY19] still computes the matrix inverse $(A^\top \operatorname{diag}(w_k)A)^{-1/2}$ exactly, resulting in an $O(nd^2)$ per-iteration cost.

## 4.2 Algorithm in Nearly Input-Sparsity Time

**Fixed Point Iteration.** Following [CCLY19], by the observation that the optimal solution $w^*$ to the program (2) satisfies $w_i^* \cdot (1 - \sigma_i(w^*)) = 0$ for all $i \in [n]$, where $\sigma_i(\cdot)$ denote the leverage score based on the constraint matrix $A$, i.e., $\sigma_i(w) := a_i^\top (A^\top \operatorname{diag}(w) A)^{-1} a_i$, where $a_i$ denote the $i$-th row vector of matrix $A$, we use the fixed point iteration method to find the John Ellipsoid. Ideally, the algorithm updates the vector $w$ by the fixed point iteration defined as:

$$
\begin{aligned}
w_{k+1,i} &= a_i^\top \sqrt{w_{k,i}} (A^\top \operatorname{diag}(w_k)A)^{-1} \sqrt{w_{k,i}} a_i \\
&= a_i^\top \sqrt{w_{k,i}} (A^\top \operatorname{diag}(w_k)A)^{-1/2} \cdot (A^\top \operatorname{diag}(w_k)A)^{-1/2} \sqrt{w_{k,i}} a_i \\
&= \|(A^\top \operatorname{diag}(w_k)A)^{-1/2} \sqrt{w_{k,i}} a_i\|_2^2
\end{aligned}
\tag{4}
$$

If we want to calculate this quantity exactly, then [CCLY19] already stated that the per iteration running time must have a dependence on quadratic dependency on $d$. Instead, we only require an *approximate* version, which comes at the cost of approximation guarantees and a failure probability. The sketching-based algorithm in [CCLY19] use a random Gaussian matrix[1] $S \in \mathbb{R}^{s \times d}$ *alone* for speedup, and the resulting update becomes $\widehat{w}_{k+1,i} := \|S(A^\top \operatorname{diag}(w_k)A)^{-1/2} \sqrt{w_{k,i}} a_i\|_2^2$.

This update mitigate the running time dependency on $d$, but still suffers a $nd^2$ running time as they calculate the inverse term exactly.

**Leverage Score Sampling.** Note that if we denote $B_k := \sqrt{\operatorname{diag}(w_k)} \cdot A$, and $b_{k,i}^\top$ is the $i$-th row of matrix $B_k$, then for $k \in [T-1]$, we can write $w_{k+1,i} = b_{k,i}^\top ((B_k)^\top B_k)^{-1} b_{k,i}$. In this light, $w_{k+1,i}$ is precisely the *leverage score* of the $i$-th row of matrix $B_k$.

To compute these leverage scores efficiently, we use *leverage score sampling* with oversampling [CLM+15, DLS23]. Specifically, if we sample rows of $B_k$ with probabilities proportional to an overestimate of their leverage scores (by a factor of $\kappa$), then with high probability, the sampled matrix provides a $(1 \pm \epsilon_0)$ approximation to $B_k^\top B_k = A^\top \operatorname{diag}(w_k)A$.

Formally, the sampling process is defined as follows.

**Definition 4.1** (Sampling process). *For any $w \in \mathbb{R}_+^n$, let $H(w) = A^\top W A$, where $W = \operatorname{diag}(w)$. Let $p_i \geq \beta \cdot \sigma_i(\sqrt{W}A)/d$, suppose we sample with replacement independently for $s$ rows of matrix $\sqrt{W}A$, with probability $p_i$ of sampling row $i$ for some $\beta \geq 1$. Let $i(j)$ denote the index of the row sampled in the $j$-th trial. Define the generated sampling matrix as*

$$\widetilde{H}(w) := \frac{1}{s} \sum_{j=1}^s \frac{1}{p_{i(j)}} w_{i(j)} a_{i(j)} a_{i(j)}^\top.$$

The following lemma provides the guarantee of the above sampling process.

---

[1] each entry draws i.i.d from a standard normal distribution $\mathcal{N}(0, 1)$

**Lemma 4.2** (Sampling using Matrix Chernoff, informal version of Lemma F.4). *Let $\epsilon_0, \delta_0 \in (0, 1)$ be the precision and failure probability parameters, respectively. Suppose $\widetilde{H}(w)$ is generated as in Definition 4.1, then with probability at least $1 - \delta_0$, we have $(1 - \epsilon_0) \cdot H(w) \preceq \widetilde{H}(w) \preceq (1 + \epsilon_0) \cdot H(w)$. Moreover, the number of rows sampled is $s = \Theta(\beta \cdot \epsilon_0^{-2} d \log(d/\delta_0))$.*

**Sketching.** In order to further speed up the algorithm, we apply sketching techniques at line 12 in Algorithm 2. For each iteration, we use a random Gaussian matrix of dimension $s \times d$ to speed up the calculation while maintaining enough accuracy.

Following all the tools above, we are able to prove the following conclusion. As shown in Algorithm 2, the algorithm first computes the iteration-averaged vector $u$ and then normalizes it to obtain the final output $v$.

**Lemma 4.3** (Approximation error, informal version of Lemma E.4). *Let $u \in \mathbb{R}^n$ denote the iteration-averaged vector computed in Algorithm 2, where $u_i = \frac{1}{T} \sum_{k=1}^{T} w_{k,i}$. Fix the number of iterations executed in the algorithm as $T = O(\epsilon^{-1} \log(n/d))$ and $s = 1000/\epsilon$. Let $\phi_i(u) := \log \sigma_i(u)$. Then for $i \in [n]$: $\phi_i(u) \leq \frac{1}{T} \log(\frac{n}{d}) + \epsilon/250 + \epsilon_0$ holds with probability $1 - \delta - \delta_0$.*

This conclusion says that, by adding the steps (line 7 to line 15 in Alg. 2) to approximate the leverage score of $B_k$, we only introduce some extra manageable failure probability and additive error terms.

### 4.3 Algorithm for Small Treewidth

Now let's move to the technical overview for the *treewidth* setting. The treewidth setting is an interesting research problem, and has been studied in many works such as [BGS21, LSZ+20, SZ23]. When the constraint matrix $A$ is an incidence matrix for a graph, it is natural to parameterize the graph in terms of its *treewidth* $\tau$.

In our second algorithm (Algorithm 3), we leverage the fact that for matrix $A$ with small treewidth $\tau$, there exist a permutation $P$ of $A$ such that the Cholesky factorization $PA^\top WAP^\top = LL^\top$ is $\tau$-sparse during the iterative algorithm, i.e., $L \in \mathbb{R}^{n \times n}$ has column sparsity $\tau$. Thus, instead of computing $B_k B_k^\top$ directly, we first decompose $B_k B_k^\top$ by $L_k L_k^\top$ in $O(n\tau^2)$ time. By using the sparsity of $L_k$, we then complete the follow-up computation of $\sigma(w)$ with $O(n\tau^2)$ time. In conclusion, we provide an implementation that takes $O((n\tau^2) \cdot T)$ to find the $(1 + \epsilon)$-approximation of John Ellipsoid.

## 5 Analysis of Input-Sparsity Algorithm

In Section 5.1, we present the running time needed for our algorithm (Algorithm 2). In Section 5.2, we provide a novel telescoping lemma. In Section 5.3, we show the correctness of our implementation.

For our discussions, especially in the context of proofs, we've also introduced some new notation to assist in comprehension and clarity. We define $Q_k := S_k H_k \in \mathbb{R}^{s \times d}$ and $\widetilde{w}_{k+1,i} := \frac{1}{s} \|Q_k \sqrt{w_{k,i}} a_i\|_2^2$.

### 5.1 Running Time of Input-Sparsity Algorithm

Next, we show the running time of Theorem 1.1.

**Lemma 5.1** (Running time of Algorithm 2, informal version of Lemma D.1). *Given a symmetric convex polytope, for all $\epsilon \in (0, 1)$, Algorithm 2 can find a $(1 + \epsilon)^2$-approximation of John Ellipsoid inside this polytope with $\epsilon_0 = \Theta(\epsilon)$ and $T = \Theta(\epsilon^{-1} \log(n/d))$ in time $\widetilde{O}((\epsilon^{-1} \operatorname{nnz}(A) + \epsilon^{-2} d^\omega) T)$.*

### 5.2 Telescoping Lemma

We introduce an innovative telescoping lemma. This stands in contrast to Lemma C.4 as mentioned in [CCLY19]. The distinction between the two is crucial: the prior telescoping lemma was restricted to sketching processes. In contrast, the lemma we are about to discuss encompasses both sketching and sampling.

At each iteration $k$ of Algorithm 2, we compute approximate weights $\widetilde{w}_{k,i}$ using sketching or sampling, introducing errors relative to exact weights $w_{k,i}$. Our telescoping analysis bounds how

---

**Algorithm 2** Faster Algorithm for approximating John Ellipsoid inside symmetric polytopes

---

1: **procedure** FASTAPPROXGENERAL($A \in \mathbb{R}^{n \times d}$)             ▷ Theorem 1.1
2:   $s \leftarrow \Theta(\epsilon^{-1})$, $T \leftarrow \epsilon^{-1} \log(n/d)$, $\epsilon_0 \leftarrow \Theta(\epsilon)$, $N \leftarrow \Theta(\epsilon_0^{-2} d \log(nd/\delta))$, $w_1 \leftarrow (d/n) \cdot \mathbf{1}_n$
3:   **for** $k = 1, \cdots, T-1$ **do**
4:    $W_k \leftarrow \mathrm{diag}(w_k)$
5:    $B_k \leftarrow \sqrt{W_k} A$ ▷ We want to compute $w_{k+1,i} = \|(B_k^\top B_k)^{-1/2}(\sqrt{w_{k,i}} a_i)\|_2^2$ by Eq. (4).
6:    Let $S_k \in \mathbb{R}^{s \times d}$ be a random matrix where each entry is chosen i.i.d from $\mathcal{N}(0,1)$
7:    Computing the $O(1)$-approximation to the leverage score of $B_k$ ▷ $\widetilde{O}(\epsilon_\sigma^{-2}(\mathrm{nnz}(A) + d^\omega))$
8:    Generate a diagonal sampling matrix $D_k \in \mathbb{R}^{n \times n}$ according to the leverage score
9:        ▷ Via matrix Chernoff, $(1 - \epsilon_0) \cdot B_k^\top B_k \preceq B_k^\top D_k B_k \preceq (1 + \epsilon_0) \cdot B_k^\top B_k$
10:    Compute $\widetilde{H}_k \leftarrow (B_k^\top D_k B_k)^{-1/2}$    ▷ Lemma 4.2, $\|D_k\|_0 = N$, $O(\epsilon_0^{-2} d^\omega \log(n/\delta))$
11:                ▷ For proof purpose, $H_k := (B_k^\top B_k)^{-1/2}$
12:    Compute $\widetilde{Q}_k \leftarrow S_k \widetilde{H}_k$          ▷ $\widetilde{Q}_k \in \mathbb{R}^{s \times d}$, $O(\epsilon^{-1} d^2)$
13:    **for** $i = 1 \rightarrow n$ **do**           ▷ $O(\epsilon^{-1} \mathrm{nnz}(A))$
14:     $\widehat{w}_{k+1,i} \leftarrow \frac{1}{s} \|\widetilde{Q}_k \sqrt{w_{k,i}} a_i\|_2^2$    ▷ $\widehat{w}_{k+1,i}$ approximates the ideal update $w_{k+1,i}$
15:    **end for**
16:    $w_{k+1} \leftarrow \widehat{w}_{k+1}$
17:   **end for**
18:   **for** $i = 1 \rightarrow n$ **do**
19:    $u_i = \frac{1}{T} \sum_{k=1}^{T} w_{k,i}$             ▷ Lemma 4.3
20:   **end for**
21:   **for** $i = 1 \rightarrow n$ **do**
22:    $v_i = \frac{d}{\sum_{j=1}^{n} u_j} u_i$              ▷ Lemma 5.3
23:   **end for**
24:   $V \leftarrow \mathrm{diag}(v)$        ▷ $V$ is a diagonal matrix with the entries of $v$
25:   **return** $V$ and $A^\top V A$
26: **end procedure**

---

these errors accumulate over $T$ iterations by decomposing the final approximation quality $\sigma_i(u)$ into two terms: an initial condition term $\frac{1}{T} \log \frac{n}{d}$ and an average per-iteration error $\frac{1}{T} \sum_{k=1}^{T} \log \frac{\widetilde{w}_{k,i}}{w_{k,i}}$. This directly motivates our choice $T = O(\epsilon^{-1} \log(n/d))$ to ensure both terms are $O(\epsilon)$, yielding $(1 + \epsilon)$-approximation.

**Lemma 5.2** (Telescoping, Algorithm 2, informal version of Lemma E.3). *Let $u \in \mathbb{R}^n$ denote the iteration-averaged vector computed in Algorithm 2, where $u_i = \frac{1}{T} \sum_{k=1}^{T} w_{k,i}$. Fix $T$ as the number of main loops executed in Algorithm 2. Let $\phi_i(u) := \log \sigma_i(u)$. Then for $i \in [n]$, $\phi_i(u) \leq \frac{1}{T} \log \frac{n}{d} + \frac{1}{T} \sum_{k=1}^{T} \log \frac{\widetilde{w}_{k,i}}{w_{k,i}} + \epsilon_0$ holds with probability $1 - \delta_0$.*

### 5.3 Correctness of Input-Sparsity Algorithm

In terms of Definition 3.3, to show Algorithm 2 provides a reasonable approximation of the John Ellipsoid, it is necessary to prove that for the output $v \in \mathbb{R}^n$ of Algorithm 2, $\sigma_i(v) \leq 1 + O(\epsilon)$, $\forall i \in [n]$. Our main result is shown below.

**Theorem 5.3** (Correctness, informal version of Theorem E.1). *Let $\epsilon_0 = \frac{\epsilon}{1000}$. Let $v \in \mathbb{R}^n$ be the output of Algorithm 2. For all $\epsilon \in (0,1)$, when $T = O(\epsilon^{-1} \log(n/d))$, we have $\Pr\left[\sigma_i(v) \leq (1+\epsilon)^2, \forall i \in [n]\right] \geq 1 - \delta - \delta_0$ Moreover, $\sum_{i=1}^{n} v_i = d$. Therefore, Algorithm 2 provides $(1+\epsilon)^2$-approximation to program Eq. (2).*

Next, we show our final result.

**Theorem 5.4** (Correctness part of Theorem 1.1). *Given a matrix $A \in \mathbb{R}^{n \times d}$, we define a centrally symmetric polytope $P$ as follows: $\{x \in \mathbb{R}^d : -\mathbf{1}_n \leq Ax \leq \mathbf{1}_n\}$. Then, given $\epsilon \in (0,1)$, Algorithm 2 that outputs an ellipsoid $Q$ satisfies: $\frac{1}{\sqrt{1+\epsilon}} \cdot Q \subseteq P \subseteq \sqrt{d} \cdot Q$.*

*Proof.* By combining Theorem 5.3 and Lemma 3.4, we can complete the proof.       □

# 6 Analysis of Small Treewidth Algorithm

In this section, we analyze the algorithm (Algorithm. 3) for constraint matrix with small treewidth (Definition 2.5). Further details are provided in Appendix G.

---

**Algorithm 3** Faster Algorithm for approximating John Ellipsoid (under tree width setting)

1: **procedure** FASTAPPROXTW($A \in \mathbb{R}^{n \times d}$)                                  ▷ Theorem 1.2
2:     $s \leftarrow \Theta(\epsilon^{-1}), T \leftarrow \Theta(\epsilon^{-1} \log(n/d)), w_1 \leftarrow (d/n) \cdot \mathbf{1}_n$
3:     **for** $k = 1, \cdots, T - 1$ **do**
4:        $W_k = \mathrm{diag}(w_k)$.
5:        $B_k = \sqrt{W_k} A$
6:        $L_k \leftarrow$ Cholesky decomposition matrix for $B_k^\top B_k$ i.e., $L_k L_k^\top = B_k^\top B_k$    ▷ $O(n\tau^2)$
7:        **for** $i = 1 \rightarrow n$ **do**
8:           $w_{k+1,i} \leftarrow b_{k,i}^\top (L_k L_k^\top)^{-1} b_{k,i}$                ▷ $O(\tau^2)$
9:        **end for**
10:     **end for**
11:     **for** $i = 1 \rightarrow n$ **do**
12:        $u_i = \frac{1}{T} \sum_{k=1}^T w_{k,i}$
13:     **end for**
14:     $U = \mathrm{diag}(u)$.             ▷ $U$ is a diagonal matrix with the entries of $u$
15:     **return** $U$ and $A^\top U A$      ▷ Approximate John Ellipsoid inside the polytope
16: **end procedure**

---

**Theorem 6.1** (Running time of Algorithm 3, informal version of Theorem G.4). *For all $\epsilon \in (0,1)$, we can find a $(1+\epsilon)$-approximation of John Ellipsoid defined by matrix $A$ with treewidth $\tau$ inside a symmetric convex polytope in time $O((n\tau^2) \cdot T)$ where $T = \epsilon^{-1} \log(n/d)$.*

*Proof sketch.* For matrices like $A$ with a small treewidth $\tau$, there exists a permutation $P$ allowing the Cholesky factorization, $PA^\top WAP^\top = LL^\top$, to be $\tau$-sparse throughout the iterative algorithm. In essence, the matrix $L$ has a column sparsity of $\tau$. Instead of directly calculating $B_k B_k^\top$, we first break down $B_k B_k^\top$ into $L_k L_k^\top$, which takes $O(n\tau^2)$ time. Utilizing the sparsity of $L_k$, the computation of $\sigma(w)$ is also achieved in $O(n\tau^2)$ time. $\qquad\square$

Next, we propose the theorem that shows the correctness of our algorithm.

**Theorem 6.2** (Correctness of Algorithm 3, informal version of Theorem G.2). *Let $u$ be the output of Algorithm 3. For all $\epsilon \in (0,1)$, when $T = O(\epsilon^{-1} \log(n/d))$, we have $\sigma_i(u) \leq (1+\epsilon)$ and $\sum_{i=1}^n u_i = d$.*

*Proof sketch.* We set $T := 1000\epsilon^{-1} \log(n/d)$ By using Corollary G.1 and the fact that for small $\epsilon$, $\epsilon/50 \leq \log(1+\epsilon)$, we have for $i \in [n]$, $\log \sigma_i(u) \leq \log(1+\epsilon)$ In conclusion, $\sigma_i(u) \leq 1 + \epsilon$. Additionally, since for $k \in [T]$, each row of $w_{k,i}$ is a leverage score of $i$-th row of matrix $B_k = \sqrt{W_k} A$, according to Lemma 2.2, we have: $\sum_{i=1}^n u_i = \sum_{i=1}^n \frac{1}{T} \sum_{k=1}^T w_{k,i} = \frac{1}{T} \sum_{k=1}^T d = d$

Thus, we complete the proof. $\qquad\square$

# 7 Conclusion

Our paper studies the problem of approximating John Ellipsoid inside a symmetric polytope, where the state-of-the-art approach [CCLY19] had a running time of $O(nd^2)$ per iteration. We proposed two fast algorithms based on different sparsity notions (i.e., number of nonzeros and treewidth) of the constraint matrix. Our first algorithm combines leverage-score-based sampling with sketching. This has allowed us to optimize the per iteration running time to $\widetilde{O}(\epsilon^{-1} \mathrm{nnz}(A) + \epsilon^{-2} d^\omega)$ with high probability, achieving logarithmic dependency on $n$. Furthermore, our second algorithm targets scenarios where the constraint matrix has a low treewidth $\tau$. By Cholesky factorization, this algorithm achieves a time complexity of $O(n\tau^2)$ per iteration.

## Acknowledgment

We thank anonymous NeurIPS reviewers for their constructive comments.

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

# Appendix

**Roadmap.** In Section A, in list some related work. In Section B, we provide some simple algebra fact. In Section C, we introduce some tools from previous work. In Section D, we give the remaining detailed proof of running time in Theorem 5.1. In Section E, we give a lemma that helps the correctness proof. In Section G, we present a faster algorithm to solve the John Ellipsoid problem with small treewidth setting. In Section F, we provide the sparsification tool used in analysis of Algorithm 2.

## A   Related Works

**Fast John Ellipsoid Computation**   There is a rich body of research on efficient algorithms for computing the John Ellipsoid. The interior point algorithm by [NN94] computes the John Ellipsoid in $O((n^{3.5} + n^{2.5}d^2)\log(n/\epsilon))$ time. [KT93] improved this to $O(n^{3.5}\log(n/\epsilon)\log(d/\epsilon))$. Subsequently, [Nem99, Ans02] developed algorithms with a time complexity of $O(n^{3.5}\log(n/\epsilon))$. The best algorithm based on convex optimization solvers, developed by [KY05, TY07], runs in $O(\epsilon^{-1}nd^3)$ time. More recently, the fixed-point iteration method by [CCLY19] achieves a time complexity of $\widetilde{O}(\epsilon^{-1}nd^2)$. For a comprehensive survey of John Ellipsoid computation, we refer readers to see [Tod16].

**Leverage score sampling**   Applying a sampling matrix for efficiency is a quite standard way in the field of numerical linear algebra (see [CW13, BWZ16, RSW16, SWZ17, SWZ19, CLS19, BLSS20, DSW22, DLS23, LSZ23, GSY23]). In our paper, we use leverage score sampling as a non-oblivious dimension reduction technique, similarly as in [SS11, BSS12, SXZ22, Zha22].

**Sketching**   Sketching is a powerful technique used in many other fundamental problems such as linear programming [JSWZ21, SY21], empirical risk minimization [LSZ19, QSZZ23], semi-definite programming [JKL+20, HJS+22, SYYZ23]. Moreover, it is a popular technique in randomized linear algebra and has been widely applied in a lot of linear algebra tasks [CW13, NN13, BWZ16, RSW16, SWZ17, XZZ18, SWZ19, LSZ19, JSWZ21, SY21, BPSW21, HSWZ22, SXYZ22, GS22, SWYZ21]. Sketching is widely applied in an oblivious way as a dimension reduction technique [CW13, NN13]. For approximate John-Ellipsoid methods, prior work [CCLY19] uses the sketching method alone, providing the potential for further optimization. [MMO22] also studied ellipsoidal approximation given a convex polytope characterized in the form of a data stream. Their problem is more challenging, and their solution is not optimal in our setting.

**Treewidth Setting**   Since the introduction of treewidth as a concept, extensive work has optimized various problems based on it. More recently, [KKMR22, GS22, GSZ23, Zha23, BGdMT23] associate treewidth with linear program solvers and enhance the efficiency of the optimization beyond matrix sparsity.

## B   Basic Tools

We provide a basic algebra claim that is used in our paper.

**Fact B.1.** *Given vector $w$, it holds that $A^\top \operatorname{diag}(w)A = \sum_{i=1}^n w_i a_i a_i^\top$, where $a_i$ is the $i$-th column of $A$.*

*Proof.* We have,

$$A^\top \operatorname{diag}(w) = [w_1 a_1, w_2 a_2, \cdots w_n a_n]$$

Then the $x, y$ element for $A^\top \operatorname{diag}(w)A$ is $\sum_{i=1}^n w_i a_{iy} a_{ix}$. Hence, $A^\top \operatorname{diag}(w)A = \sum_{i=1}^n w_i a_i a_i^\top$. $\square$

We introduce some facts that are useful to our proof.

**Fact B.2.** *For any real numbers $a \geq 1$ and $b \geq 2$, we have*

$$\log(ab) \leq 2a \cdot \log b$$

*Proof.* We have

$$
\begin{aligned}
\log(ab) &\leq \log a + \log b \\
&\leq a + \log b \\
&\leq a \log b + \log b \\
&\leq a \log b + a \log b \\
&\leq 2a \log b.
\end{aligned}
$$

where the third step follows from $\log b \geq 1$, the forth step follows from $a \geq 1$.

Thus, we complete the proof. $\square$

**Fact B.3.** *For any $a \geq 1$ and $b \geq 2$, we have*
$$
a + \log(ab) \leq 3a \log b
$$

*Proof.* Using Fact B.2, we have
$$
\log(ab) \leq 2a \log b
$$
Then we have
$$
a + \log(ab) \leq a + 2a \log b \leq 3a \log b
$$
where the last step follows from $a \leq a \log b$. $\square$

**Fact B.4.** *For any $n, d$ such that $2 \leq d \leq n \leq \mathrm{poly}(d)$. For any $\delta \in (0, 0.1)$, we have*
$$
\log(d \log(n/d)/\delta) = O(\log(d/\delta))
$$

*Proof.* Let $c > 1$ denote some constant value such that $n \leq d^c$.

Then we can write
$$
\begin{aligned}
d \log(n/d) &\leq d \log(d^{c-1}) \\
&= (c-1)d \log d \\
&\leq cd \log d \\
&\leq cd^2
\end{aligned}
$$
where the first step follows from $n \leq d^c$, and the last step follows from $\log d \leq d$.

Thus
$$
\begin{aligned}
\log(d \log(n/d)/\delta) &\leq \log(cd^2/\delta) \\
&\leq 2c \log(d^2/\delta) \\
&\leq 2c \log(d^2/\delta^2) \\
&= 4c \log(d/\delta) \\
&= O(\log(d/\delta)).
\end{aligned}
$$
where the second step follows from Fact B.2, the third step follows from $\delta \in (0, 1)$. $\square$

# C    Tools From Previous Works

We provide a bounding expectation in Section C.1 and show the convexity in Section C.2.

## C.1    Bounding expectation

**Lemma C.1** (Implicitly in Lemma C.5 and Lemma C.6 in arXiv[2] version of [CCLY19]). *If $s$ is even, define $\lambda_i(w_k) = \log \frac{\widetilde{w}_{k,i}}{w_{k,i}}$ then we have*
$$
\mathbb{E}[\lambda_i(w_k)] = \frac{2}{s}
$$
$$
\mathbb{E}[(\exp(\lambda_i(w_k)))^\alpha] \leq (\frac{n}{d})^{\frac{\alpha}{T}} \cdot (1 + \frac{2\alpha}{sT - 2\alpha})^T.
$$
*where the randomness is taken over the sketching matrices $\{S^{(k)}\}_{k=1}^{T-1}$.*

[2]https://arxiv.org/pdf/1905.11580.pdf

## C.2 Convexity

Here, we show the convexity of $\phi_i$.

**Lemma C.2** (Convexity, Lemma 3.4 in arXiv [CCLY19]). *For $i = 1, \cdots, n$, let $\phi_i : \mathbb{R}^n \to \mathbb{R}$ be the function defined as*

$$\phi_i(v) = \log \sigma_i(v) = \log(a_i^\top (\sum_{j=1}^n v_j a_j a_j^\top)^{-1} a_i).$$

*Then $\phi_i$ is convex.*

## D Proofs of Running Time of Input-Sparsity Algorithm

**Lemma D.1** (Performance of Algorithm 2, formal version of Lemma 5.1). *Given a symmetric convex polytope, for all $\epsilon \in (0, 1)$, Algorithm 2 can find a $(1 + \epsilon)^2$-approximation of John Ellipsoid inside this polytope with $\epsilon_0 = \Theta(\epsilon)$ and $T = O(\epsilon^{-1} \log(n/d))$ in time*

$$O((\epsilon^{-1} \log(d/\delta) \cdot \mathrm{nnz}(A) + \epsilon^{-2} \log(n/\delta) \cdot d^\omega)T),$$

*where $\omega \approx 2.37$ denote the current matrix multiplication exponent [Wil12, LG14, AW21, DWZ22, ADW+24].*

*Proof.* At first, initializing the vector $w \in \mathbb{R}^n$ takes $O(n)$ time. In the main loop, the per iteration running time can be decomposed as follows:

- Calculating matrix $B_k \in \mathbb{R}^{n \times d}$ takes $O(\mathrm{nnz}(A))$ time. Due to the structure of matrix $W_k$, we only need to multiply the non-zero entries of $i$-th row by $w_{k,i}$ to get matrix $B_k$. The total non-zero entries here is $\mathrm{nnz}(A)$.

- Initializing matrix $S_k \in \mathbb{R}^{s \times d}$, where $s = \Theta(\epsilon^{-1})$, takes $O(\epsilon^{-1}n)$ time.

- Generating diagonal matrix $D_k \in \mathbb{R}^{n \times n}$ takes $\widetilde{O}(\epsilon_\sigma^{-2}(\mathrm{nnz}(A) + d^\omega))$ time by using Lemma 2.3.

- Computing matrix $\widetilde{H}_k = (B_k^\top D_k B_k)^{-1/2}$ contains three steps.

  - We first compute $B_k^\top D_k B_k \in \mathbb{R}^{d \times d}$, where $D_k$ is a diagonal matrix with $N$ non-zero entries and $N = \Theta(\epsilon_0^{-2} d \log(nd/\delta))$. It takes $O(\epsilon_0^{-2} d^\omega \log(nd/\delta))$ time by using fast matrix multiplication. As $n = \mathrm{poly}(d)$, we can simplify it as

    $$O(\epsilon_0^{-2} d^\omega \log(n/\delta)).$$

  - Second, we compute the inverse of the result in the first step, which takes $O(d^\omega)$ time
  - Third, we take the square root of the result in second step. To take square root of a matrix $T \in \mathbb{R}^{d \times d}$, we can first decompose $T$ as $U\Sigma V^\top$ using SVD, which takes $O(d^\omega)$. Then we take the square root of the diagonal matrix $\Sigma$, which takes $O(d)$. Then, we multiply them back together to get $T^{1/2}$, which takes $O(d^\omega)$. Hence, the time needed for the final step is

    $$O(d^\omega) + O(d) + O(d^\omega) = O(d^\omega)$$

  As $O(d^\omega)$ is less than $O(\epsilon_0^{-2} d^\omega \log(n/\delta))$, the total running time for computing $\widetilde{H}_k$ is $O(\epsilon_0^{-2} d^\omega \log(n/\delta))$.

- Computing matrix $\widetilde{Q}_k$ takes $O(\epsilon^{-1} d^2)$ time.

- Updating vector $w_{k+1}$ takes $O(\epsilon^{-1} \mathrm{nnz}(A))$ time. We need $O(\epsilon^{-1} \mathrm{nnz}(a_i))$ time for each iteration to compute $\frac{1}{s} \|\widetilde{Q}_k \sqrt{w_{k,i}} a_i\|_2^2$. Hence to update vector $w_{k+1}$, we need

  $$\sum_{i=1}^n O(\epsilon^{-1} \mathrm{nnz}(a_i)) = O(\epsilon^{-1} \mathrm{nnz}(A))$$

  time.

In summary, the overall per iteration running time for the main loop is

$$O(\epsilon^{-1} \log(d/\delta) \cdot \mathrm{nnz}(A) + \epsilon^{-2} \log(n/\delta) \cdot d^\omega)$$

where

$$\epsilon_\sigma = \Theta(1) \text{ and } \delta_\sigma = \frac{\delta}{T} = \frac{\delta\epsilon}{\log(n/d)}$$

Hence, with $\epsilon_\sigma = \Theta(1)$ and $\delta_\sigma = \frac{\delta}{T} = \frac{\delta\epsilon}{\log(n/d)}$, the overall per iteration running time for the main loop is

$$
\begin{aligned}
&O(\mathrm{nnz}(A)) + O(\epsilon^{-1}n) + \widetilde{O}(\epsilon_\sigma^{-2}(\mathrm{nnz}(A) + d^\omega)) + O(\epsilon_0^{-2}d^\omega \log(n/\delta)) + O(\epsilon^{-1}d^2) + O(\epsilon^{-1} \mathrm{nnz}(A)) \\
&= \widetilde{O}(\epsilon_\sigma^{-2}(\mathrm{nnz}(A) + d^\omega)) + O(\epsilon_0^{-2}d^\omega \log(n/\delta)) + O(\epsilon^{-1}d^2) + O(\epsilon^{-1} \mathrm{nnz}(A)) \\
&= O(\epsilon_\sigma^{-2}(\mathrm{nnz}(A) + d^\omega) \log(d/\delta_\sigma) + \epsilon_0^{-2}d^\omega \log(n/\delta) + \epsilon^{-1}d^2 + \epsilon^{-1} \mathrm{nnz}(A)) \\
&= O((\epsilon_\sigma^{-2} \log(d/\delta_\sigma) + \epsilon^{-1}) \mathrm{nnz}(A) + (\epsilon_\sigma^{-2} \log(d/\delta_\sigma) + \epsilon_0^{-2} \log(n/\delta))d^\omega + \epsilon^{-1}d^2) \\
&= O((\log(d/\delta_\sigma) + \epsilon^{-1}) \mathrm{nnz}(A) + (\log(d/\delta_\sigma) + \epsilon_0^{-2} \log(n/\delta))d^\omega + \epsilon^{-1}d^2)
\end{aligned}
$$

where the first step comes from $\mathrm{nnz}(A) > n$ and $\mathrm{nnz}(A) > d$, the second step follows from the definition of $\widetilde{O}$, the third step follows from reorganization, the fourth step follows from $\epsilon_\sigma = \Theta(1)$.

Note that without loss of generality, we can assume $2 \le d \le n \le \mathrm{poly}(d)$. For convenient of the simplifying complexity related to logs, we can assume $n \ge 2d$ and $\delta \in (0, 0.1)$ and $\epsilon \in (0, 0.1)$.

We can try to further simplify $\log(d/\delta_\sigma) + \epsilon^{-1}$, using the definition of $\delta_\sigma = \frac{\delta}{T} = \frac{\delta\epsilon}{\log(n/d)}$, then we can have

$$
\begin{aligned}
\log(d/\delta_\sigma) + \epsilon^{-1} &= \log(\frac{d \log(n/d)}{\delta\epsilon}) + \epsilon^{-1} \\
&= O(\epsilon^{-1} \log(\frac{d \log(n/d)}{\delta})) \\
&= O(\epsilon^{-1} \log(d/\delta))
\end{aligned}
$$

where the first step follows from definition of $\delta_\sigma$, the second step follows from Fact B.3 and the last step follows from Fact B.4.

Hence yields the total running time for the main loop as

$$O((\epsilon^{-1} \log(d/\delta) \cdot \mathrm{nnz}(A) + (\log(d/(\delta\epsilon)) + \epsilon_0^{-2} \log(n/\delta)) \cdot d^\omega + \epsilon^{-1} \cdot d^2)T).$$

Then, computing the average of vector $w$ from time $1$ to $T$, and computing the vector $v_i$ takes $O(nT)$ time. Finally, note that we don't have to output $A^\top V A$. Instead, we can just output $A$ and vector $v$, which takes $O(n)$ time.

Therefore, by calculation, the running time of Algorithm 2 is:

$$
\begin{aligned}
&O((\epsilon^{-1} \log(d/\delta) \cdot \mathrm{nnz}(A) + (\log(d/(\delta\epsilon)) + \epsilon_0^{-2} \log(n/\delta)) \cdot d^\omega + \epsilon^{-1} \cdot d^2)T) \\
&= O((\epsilon^{-1} \log(d/\delta) \cdot \mathrm{nnz}(A) + (\log(d/(\delta\epsilon)) + \epsilon^{-2} \log(n/\delta)) \cdot d^\omega)T) \\
&= O((\epsilon^{-1} \log(d/\delta) \cdot \mathrm{nnz}(A) + \epsilon^{-2} \log(n/\delta) \cdot d^\omega)T)
\end{aligned}
$$

where the first step comes from $\epsilon_0 = \Theta(\epsilon)$ and $\omega \ge 2$, and the last step follows from $n > d$ and $\epsilon \in (0, 1)$. Note that $\omega$ denotes the exponent of matrix multiplication [Wil12, LG14, AW21]. $\square$

## E Proofs Of Correctness of Input-Sparsity Algorithm

In Section E.1, we show that Algorithm 2 gives a reasonable approximation of the John Ellipsoid. In Section E.2, we provide the bound of $\lambda_i$. In Section E.3, we provide the formal version of telescoping. In Section E.4, we give the upper bound of $\phi_i$.

## E.1 Main Result

**Theorem E.1** (Correctness, restatement of Theorem 5.3). *Let $\epsilon_0 = \frac{\epsilon}{1000}$. Let $v \in \mathbb{R}^n$ be the output of Algorithm 2. For all $\epsilon \in (0,1)$, when $T = O(\epsilon^{-1} \log(n/d))$, we have*

$$\Pr\left[\sigma_i(v) \leq (1+\epsilon)^2, \forall i \in [n]\right] \geq 1 - \delta - \delta_0$$

*Moreover,*

$$\sum_{i=1}^{n} v_i = d.$$

*Therefore, Algorithm 2 provides $(1+\epsilon)^2$-approximation to program Eq. (2)*

*Proof.* We set

$$T = 1000\epsilon^{-1} \log(n/d) \quad \text{and} \quad \epsilon_0 = \epsilon/1000,$$

By Lemma E.4, we know the succeed probability is $1 - \delta - \delta_0$. Then, we have for $i \in [n]$,

$$\log \sigma_i(u) = \phi_i(u)$$
$$\leq \frac{1}{T} \log(n/d) + \epsilon/250 + \epsilon_0$$
$$\leq \frac{\epsilon}{50}$$
$$\leq \log(1+\epsilon)$$

where the first step uses the definition of $\sigma_i$, the second step uses Lemma 4.3, the third step comes from calculation, and the last step comes from the fact that when $0 < \epsilon < 1$, $\frac{\epsilon}{50} \leq \log(1+\epsilon)$.

In conclusion, $\sigma_i(u) \leq 1 + \epsilon$.

Because, we choose $v_i = \frac{d}{\sum_{j=1}^{n} u_j} u_i$, then $\sum_{i=1}^{n} v_i = d$.

Next, we have

$$\sigma_i(v) = a_i^\top (A^\top V A)^{-1} a_i$$
$$= a_i^\top \left(\frac{d}{\sum_{i=1}^{n} u_i} A^\top U A\right)^{-1} a_i$$
$$= \frac{\sum_{i=1}^{n} u_i}{d} \sigma_i(u)$$
$$\leq (1+\epsilon) \cdot \sigma_i(u)$$
$$\leq (1+\epsilon) \cdot (1+\epsilon)$$

where the first step uses the definition of $\sigma_i(v)$, the second step uses the definition of $V$, the third step uses the definition of $\sigma_i(u)$, the fourth step comes from $u_i$ is at most $(1+\epsilon)$ true leverage score, and the summation of true leverage scores is $d$ (by Lemma 2.2), the last step comes from $\sigma_i(u) \leq (1+\epsilon)$.

Thus, we complete the proof. $\square$

## E.2 High Probability Bound of $\lambda_i$

We provide a high probability bound of $\lambda_i$ as follows.

**Lemma E.2** (High probability Argument on $\lambda_i(w)$). *Let $\lambda_i(w) = \log \frac{\widetilde{w}_{k,i}}{w_{k,i}}$. Then we have*

$$\Pr[\exp(\lambda_i(w)) \geq 1 + \epsilon] \leq \frac{\left(\frac{n}{d}\right)^{\frac{\alpha}{T}} e^{\frac{4\alpha}{s}}}{(1+\epsilon)^\alpha}.$$

*Moreover, with our choice of $s, T$, with large enough $n$ and $d$, we have:*

$$\Pr[\exp(\lambda_i(w)) \geq 1 + \epsilon] \leq \frac{\delta}{n}$$

*Proof.* In the proof, we pick $\alpha = \frac{2}{\epsilon} \log \frac{n}{\delta}$. By the choice of $\alpha$, we have that:

$$\alpha \geq \frac{\log(n/\delta)}{\log \frac{1+\epsilon}{1+\epsilon/4}} \tag{5}$$

$$sT \geq 4\alpha \tag{6}$$

Then, for $i \in [n]$, by Markov Inequality on the $\alpha$ moment of $\exp(\lambda_i(w))$, we have that:

$$\Pr[\exp(\lambda_i(w)) \geq 1 + \epsilon] = \Pr[\exp(\lambda_i(w))^\alpha \geq (1+\epsilon)^\alpha]$$
$$\leq \frac{\mathbb{E}[\exp(\lambda_i(w))^\alpha]}{(1+\epsilon)^\alpha}$$
$$\leq \frac{(\frac{n}{d})^{\frac{\alpha}{T}} \cdot (1 + \frac{2\alpha}{sT-2\alpha})^T}{(1+\epsilon)^\alpha}$$
$$\leq \frac{(\frac{n}{d})^{\frac{\alpha}{T}} \cdot (1 + \frac{2\alpha}{sT/2})^T}{(1+\epsilon)^\alpha}$$
$$\leq \frac{\frac{n}{d}^{\frac{\alpha}{T}} e^{\frac{4\alpha}{s}}}{(1+\epsilon)^\alpha}$$

where the first step comes from calculation, the second step comes from Markov Inequality, the third step comes from applying Lemma C.1, the fourth step comes from the choice of $\alpha$ that $sT \geq 4\alpha$, and the final step comes from $1 + x \leq e^x$.

Moreover, for large enough $n$ and $d$, we have that:

$$(\frac{n}{d})^{\frac{1}{T}} = (\frac{n}{d})^{\frac{\epsilon/10}{\log(n\delta)}} \leq 1 + \epsilon/10 \tag{7}$$

Also, we have:

$$e^{\frac{4}{s}} = e^{\frac{\epsilon}{20}} \leq 1 + \epsilon/10 \tag{8}$$

Hence,

$$\Pr[\exp(\lambda_i(w)) \geq 1 + \epsilon] \leq (\frac{(1+\epsilon/10)^2}{1+\epsilon})^\alpha$$
$$\leq (\frac{1+\epsilon/4}{1+\epsilon})^\alpha$$
$$\leq \frac{\delta}{n}$$

where the first step comes from applying Eq (7) and Eq. (8), the second step comes from calculation, and the last step comes from Eq. (5). $\square$

### E.3 Proof of Lemma 5.2

**Lemma E.3** (Telescoping, Algorithm 2, restatement of Lemma 5.2). *Fix $T$ as the number of main loops executed in Algorithm 2. Let $u \in \mathbb{R}^n$ denote the iteration-averaged vector computed in Algorithm 2, where $u_i = \frac{1}{T}\sum_{k=1}^T w_{k,i}$. Then for $i \in [n]$, with probability $1 - \delta_0$,*

$$\phi_i(u) \leq \frac{1}{T} \log \frac{n}{d} + \frac{1}{T} \sum_{k=1}^T \log \frac{\widetilde{w}_{k,i}}{w_{k,i}} + \epsilon_0$$

*Proof.* We define

$$u := (u_1, u_2, \cdots, u_n) \in \mathbb{R}^n.$$

For $k = 1, \cdots, T - 1$, we define

$$w_k := (w_{k,1}, \cdots, w_{k,n}) \in \mathbb{R}^n$$

and

$$\widehat{w}_{k+1} := (w_{k,1} \cdot \sigma_1(w_k), \cdots, w_{k,n} \cdot \sigma_n(w_k)).$$

By the convexity of $\phi_i$ (Lemma C.2)

$$\phi_i(u) = \phi_i(\frac{1}{T} \sum_{k=1}^{T} w_k)$$

$$\leq \frac{1}{T} \sum_{k=1}^{T} \phi_i(w_k)$$

$$= \frac{1}{T} \sum_{k=1}^{T} \log \sigma_i(w_k)$$

$$= \frac{1}{T} \sum_{k=1}^{T} \log \frac{\widehat{w}_{k+1,i}}{w_{k,i}}$$

$$= \frac{1}{T} \sum_{k=1}^{T} \log \frac{\widehat{w}_{k+1,i} \cdot \widehat{w}_{k,i} \cdot \widetilde{w}_{k,i}}{\widehat{w}_{k,i} \cdot \widetilde{w}_{k,i} \cdot w_{k,i}}$$

$$= \frac{1}{T} (\sum_{k=1}^{T} \log \frac{\widehat{w}_{k+1,i}}{\widehat{w}_{k,i}} + \sum_{k=1}^{T} \log \frac{\widehat{w}_{k,i}}{\widetilde{w}_{k,i}} + \sum_{k=1}^{T} \log \frac{\widetilde{w}_{k,i}}{w_{k,i}})$$

$$= \frac{1}{T} \log \frac{\widehat{w}_{T+1,i}}{\widehat{w}_{1,i}} + \frac{1}{T} \sum_{k=1}^{T} \log(\frac{\widehat{w}_{k,i}}{\widetilde{w}_{k,i}}) + \frac{1}{T} \sum_{k=1}^{T} \log \frac{\widetilde{w}_{k,i}}{w_{k,i}}$$

$$= \frac{1}{T} \log \frac{n\widehat{w}_{T+1,i}}{d} + \frac{1}{T} \sum_{k=1}^{T} \log(\frac{\widehat{w}_{k,i}}{\widetilde{w}_{k,i}}) + \frac{1}{T} \sum_{k=1}^{T} \log \frac{\widetilde{w}_{k,i}}{w_{k,i}}$$

$$\leq \frac{1}{T} \log \frac{n}{d} + \frac{1}{T} \sum_{k=1}^{T} \log(\frac{\widehat{w}_{k,i}}{\widetilde{w}_{k,i}}) + \frac{1}{T} \sum_{k=1}^{T} \log \frac{\widetilde{w}_{k,i}}{w_{k,i}}$$

$$\leq \frac{1}{T} \log \frac{n}{d} + \log(1 + \epsilon_0) + \frac{1}{T} \sum_{k=1}^{T} \log \frac{\widetilde{w}_{k,i}}{w_{k,i}}$$

$$\leq \frac{1}{T} \log \frac{n}{d} + \epsilon_0 + \frac{1}{T} \sum_{k=1}^{T} \log \frac{\widetilde{w}_{k,i}}{w_{k,i}}$$

where the first step uses the definition of $u$, the second step uses the convexity of $\phi_i$, the third step uses the definition of $\phi_i$, the fourth step uses the definition of $\sigma_i$, the fifth step comes from reorganization, the sixth step comes from reorganization, the seventh step comes from reorganization, the eighth step uses our initialization on $w_1$, the ninth step comes from Lemma 2.2, the tenth step uses Corollary F.5, and the final step comes from the fact $\log(1 + \epsilon_0) \leq \epsilon_0$.

Note that, the tenth step only holds with probability $1 - \delta_0$, which gives us the high probability argument in the lemma statement.

$\square$

## E.4    Upper Bound of $\phi_i$

Then, we show the upper bound of $\phi_i$.

**Lemma E.4** ($\phi_i$, formal version of Lemma 4.3)**.** *Let $u$ be the vector generated during the Algorithm 2, fix the number of iterations executed in the algorithm as $T$ and $s = 1000/\epsilon$, with $1 - \delta - \delta_0$, we have*

$$\phi_i(u) \leq \frac{1}{T} \log(\frac{n}{d}) + \epsilon/250 + \epsilon_0 \quad \forall i \in [n].$$

*Proof.* To begin with, by Lemma 5.2, we have that, with probability $1 - \delta_0$,

$$\phi_i(u) \leq \frac{1}{T} \log \frac{n}{d} + \frac{1}{T} \sum_{k=1}^{T} \log \frac{\widetilde{w}_{k,i}}{w_{k,i}} + \epsilon_0$$

$$= \frac{1}{T} \log \frac{n}{d} + \frac{1}{T} \sum_{k=1}^{T} \lambda_i(w_k) + \epsilon_0$$

We have with probability $1 - \delta - \delta_0$, for all $i \in [n]$:

$$\phi_i(u) \leq \frac{1}{T} \log \frac{n}{d} + \epsilon/1000 + \epsilon_0$$

$$\leq \frac{1}{T} \log \frac{n}{d} + \frac{\epsilon}{250} + \epsilon_0.$$

where the first step follows from Lemma E.2. $\qquad\square$

# F   Sampling

In this section, we provide the sparsification tool used in Line 7 of Algorithm 2. Especially, we show how to approximate the matrix that has pattern $A^\top W A$, where $W$ is some non-negative diagonal matrix, by using sample matrix $D$.

**Lemma F.1** (Matrix Chernoff Bound [Tro11]). *Let $X_1, \ldots, X_s$ be i.i.d. symmetric random matrices with $\mathbb{E}[X_1] = 0$, $\|X_1\| \leq \gamma$ almost surely and $\|\mathbb{E}[X_1^\top X_1]\| \leq \sigma^2$. Let $C = \frac{1}{s} \sum_{i \in [s]} X_i$. For any $\epsilon \in (0, 1)$, it holds that*

$$\Pr[\|C\| \geq \epsilon] \leq 2d \cdot \exp\left(-\frac{s\epsilon^2}{\sigma^2 + \gamma\epsilon/3}\right).$$

To better monitor the whole process, it is useful to write $H(w)$ as $A^\top W A$, where $A \in \mathbb{R}^{n \times d}$ is the constraint matrix and $W$ is a diagonal matrix with $W = \mathrm{diag}(w)$. The sparsification process is then sample the rows from the matrix $\sqrt{W} A$.

We define the leverage score as follows:

**Definition F.2.** *Let $B \in \mathbb{R}^{n \times d}$ be a full rank matrix. We define the leverage score of the $i$-th row of $B$ as*

$$\sigma_i(B) := b_i^\top (B^\top B)^{-1} b_i,$$

*where $b_i$ is the $i$-th row of $B$.*

Next we define our sampling process as follows:

**Definition F.3** (Sampling process). *For any $w \in K$, let $H(w) = A^\top W A$. Let $p_i \geq \beta \cdot \sigma_i(\sqrt{W} A)/d$, suppose we sample with replacement independently for $s$ rows of matrix $\sqrt{W} A$, with probability $p_i$ of sampling row $i$ for some $\beta \geq 1$. Let $i(j)$ denote the index of the row sampled in the $j$-th trial. Define the generated sampling matrix as*

$$\widetilde{H}(w) := \frac{1}{s} \sum_{j=1}^{s} \frac{1}{p_{i(j)}} w_{i(j)} a_{i(j)} a_{i(j)}^\top.$$

For our sampling process defined as Definition F.3, we can have the following guarantees:

**Lemma F.4** (Sampling using Matrix Chernoff, formal version of Lemma 4.2). *Let $\epsilon_0, \delta_0 \in (0, 1)$ be the precision and failure probability parameters, respectively. Suppose $\widetilde{H}(w)$ is generated as in Definition F.3, then with probability at least $1 - \delta_0$, we have*

$$(1 - \epsilon_0) \cdot H(w) \preceq \widetilde{H}(w) \preceq (1 + \epsilon_0) \cdot H(w).$$

*Moreover, the number of rows sampled is*

$$s = \Theta(\beta \cdot \epsilon_0^{-2} d \log(d/\delta_0)).$$

*Proof.* The proof follows from the high level idea of Lemma 5.2 in [DSW22] by designing the family of random matrices $X$. Let

$$y_i = (A^\top W A)^{-1/2} \sqrt{W}_{i,i} \cdot a_i$$

be the $i$-th sampled row and set $Y_i = \frac{1}{p_i} y_i y_i^\top$.

Using $H(w) = A^\top W A$, we can write

$$y_i = (H(w))^{-1/2} \sqrt{W}_{i,i} \cdot a_i.$$

Let $X_i = Y_i - I_d$. Note that

$$
\begin{aligned}
& \sum_{i=1}^n y_i y_i^\top \\
= {} & \sum_{i=1}^n H(w)^{-1/2} W_{i,i} \cdot a_i a_i^\top H(w)^{-1/2} \\
= {} & H(w)^{-1/2} \left( \sum_{i=1}^n W_{i,i} a_i a_i^\top \right) H(w)^{-1/2} \\
= {} & H(w)^{-1/2} (A^\top W A) H(w)^{-1/2} \\
= {} & I_d.
\end{aligned}
\tag{9}
$$

where the first step uses the definition of $y_i$, the second step comes from reorganization, the third step comes from Fact B.1, and the last step uses the definition of $H(w)$.

Also, the norm of $y_i$ connects directly to the leverage score:

$$
\begin{aligned}
\|y_i\|_2^2 &= \sqrt{W}_{i,i} a_i^\top (A^\top W A)^{-1} \sqrt{W}_{i,i} a_i \\
&= \sigma_i(\sqrt{W} A).
\end{aligned}
\tag{10}
$$

We use $i(j)$ to denote the index of row that has been sampled during $j$-th trial.

We first show that $\mathbb{E}[X] = 0$. Note that

$$
\begin{aligned}
\mathbb{E}[X] &= \mathbb{E}[Y] - I_d \\
&= \left( \sum_{i=1}^n p_i \cdot \frac{1}{p_i} y_i y_i^\top \right) - I_d \\
&= 0.
\end{aligned}
$$

where the first step uses the definition of $X$, the second step uses the definition of $Y$ and the definition of expectation, and the last step uses Eq. (9).

Now, to bound $\|X\|$, we provide a bound for any $\|X_i\|$ as follows

$$
\begin{aligned}
\|X_i\| &= \|Y_i - I_d\| \\
&\le 1 + \|Y_i\| \\
&= 1 + \frac{\|y_i y_i^\top\|}{p_i} \\
&\le 1 + \frac{d \cdot \|y_i\|_2^2}{\beta \cdot \sigma_i(\sqrt{W} A)} \\
&= 1 + \frac{d}{\beta}.
\end{aligned}
$$

where the first step uses the definition of $X_i$, the second step uses triangle inequality and the definition of $I_d$, the third step uses the definition of $Y_i$, the fourth step comes from $p_i \ge \beta \cdot \sigma_i(\sqrt{W} A)/d$ and the definition of $\ell_2$ norm and the last step comes from Eq. (10).

Then we bound $\| \mathbb{E}[X^\top X] \|$ as follows.

$$\mathbb{E}[X^\top X]$$
$$= \mathbb{E}[I_d^2] + \mathbb{E}[Y^\top Y] - 2\,\mathbb{E}[Y]$$
$$= I_d + \sum_{i=1}^n p_i \frac{y_i^\top y_i y_i y_i^\top}{p_i^2} - 2\sum_{i=1}^n p_i \frac{y_i y_i^\top}{p_i}$$
$$= I_d + \left(\sum_{i=1}^n \frac{\sigma_i(\sqrt{W}A)}{p_i} y_i y_i^\top\right) - 2I_d$$
$$\leq \sum_{i=1}^n \frac{d}{\beta} y_i y_i^\top - I_d$$
$$= \left(\frac{d}{\beta} - 1\right)I_d,$$

where the first step uses definition of $X$, the second step uses the definition of $Y$ and the definition of expectation, the third step follows from Eq. (9), Eq. (10) and the definition of expectation, the third step comes from $p_i \geq \beta \cdot \sigma_i(\sqrt{W}A)/d$, and the last step comes from Eq. (9) and distributive property.

The spectral norm is then

$$\| \mathbb{E}[X^\top X] \| \leq \frac{d}{\beta} - 1.$$

Putting everything together, we choose

$$\gamma = 1 + \frac{d}{\beta}, \quad \sigma^2 = \frac{d}{\beta} - 1$$

and then we apply Matrix Chernoff Bound as in Lemma F.1:

$$\Pr[\|C\| \geq \epsilon_0]$$
$$\leq 2d \cdot \exp\left(-\frac{s\epsilon_0^2}{d/\beta - 1 + (1 + d/\beta)\epsilon_0/3}\right)$$
$$= 2d \cdot \exp(-s\epsilon_0^2 \cdot \Theta(\beta/d))$$
$$\leq \delta_0$$

where we choose $s = \Theta(\beta \cdot \epsilon_0^{-2} d \log(d/\delta_0))$.

Finally, we can show that

$$C = \frac{1}{s}\left(\sum_{j=1}^s \frac{1}{p_{i(j)}} y_{i(j)} y_{i(j)}^\top - I_d\right)$$
$$= H(w)^{-1/2}\left(\frac{1}{s}\sum_{j=1}^s \frac{1}{p_{i(j)}} w_{i(j)} a_{i(j)} a_{i(j)}^\top\right)H(w)^{-1/2} - I_d$$
$$= H(w)^{-1/2}\widetilde{H}(w)H(w)^{-1/2} - I_d.$$

where the first step uses the definition of $C$, the second step uses the definition of $y_{i(j)}$, and the last step uses the definition of $\widetilde{H}(w)$.

Therefore, we can conclude the desired result via $\|C\| \geq \epsilon_0$. $\qquad\square$

**Corollary F.5.** *Let $\epsilon_0$ denote the parameter defined as Algorithm 2. Then we have with probability $1 - \delta_0$*

$$(1 - \epsilon_0) \cdot \widetilde{w}_i \leq \widehat{w}_i \leq (1 + \epsilon_0)\widetilde{w}_i,$$

*for all $i \in [n]$.*

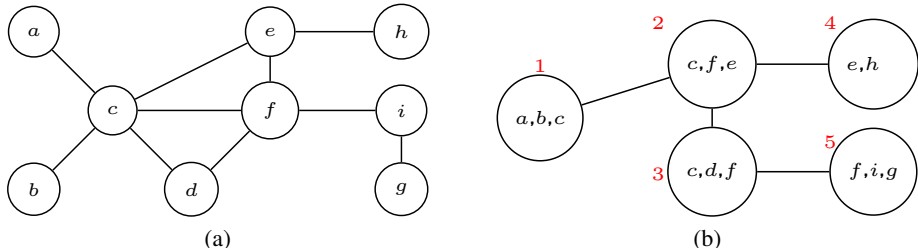

Figure 3: (a) A graph $G(V, E)$ (b) The tree decomposition for graph $G$. We can see that the union of the vertices in all bags are nodes $a, \cdots, i$, which is the same as $V(G)$. For every edge $u, v \in V(G)$, we can find at least one bag containing $u$ and $v$. For example, for edge $(c, b)$ in graph $G$, bag 1 contains both $c$ and $b$. Furthermore, the bags containing any one node in $(a)$ is a subgraph of tree $(b)$. For example, the bags containing node $c$ are bags $1, 2, 3$, which is a subgraph of the tree. Similarly, we can see that the bags containing node $f$ is bags $3, 5$, which is also a subgraph of the tree. For edge $(c, f)$, bag 2 and 3 both contain vertices $c$ and $f$. For edge $(i, g)$, bag 5 contains vertices $i$ and $g$.

*Proof.* Since if

$$(1 - \epsilon_0)A \preceq B \preceq (1 + \epsilon_0)A,$$

then for all $x$, we know

$$(1 - \epsilon_0) \cdot x^\top A x \leq x^\top B x \leq (1 + \epsilon_0) \cdot x^\top A x.$$

Thus, using lemma (Lemma F.4) implies the weights guarantees. □

## G    Small Treewidth Setting

In this section, we provide an algorithm (Algorithm 3) that approximate the John Ellipsoid in $O(\epsilon^{-1} \cdot (n\tau^2) \cdot \log(n/d))$ time with small treewidth setting. In Section G.1, we prove the correctness of our implementation. In Section G.2, we show the running time of it.

### G.1    Correctness

Note that for Algorithm 3, we compute the exact leverage score of each row, the randomness of sketching matrix $S$ and diagonal sampling $D$ doesn't play a role in our analysis. It immediately follows that the following corollary holds:

**Corollary G.1** (Telescoping, Algorithm 3). *Fix $T$ as the number of main loops executed in Algorithm 3. Let $u \in \mathbb{R}^n$ denote the iteration-averaged vector computed in Algorithm 3, where $u_i = \frac{1}{T} \sum_{k=1}^{T} w_{k,i}$. Then for $i \in [n]$,*

$$\phi_i(u) \leq \frac{1}{T} \log \frac{n}{d}$$

Next, we prove the correctness of our implementation with small treewidth setting.

**Theorem G.2** (Correctness of Algorithm 3, formal version of Theorem 6.2). *Let $u$ be the output of Algorithm 3. For all $\epsilon \in (0, 1)$, when $T = O(\epsilon^{-1} \log(n/d))$, we have:*

$$\sigma_i(u) \leq (1 + \epsilon)$$

$$\sum_{i=1}^{n} u_i = d$$

*Proof.* We set

$$T = 1000\epsilon^{-1} \log(n/d)$$

We also have for $i \in [n]$,

$$\log \sigma_i(u) = \phi_i(u)$$

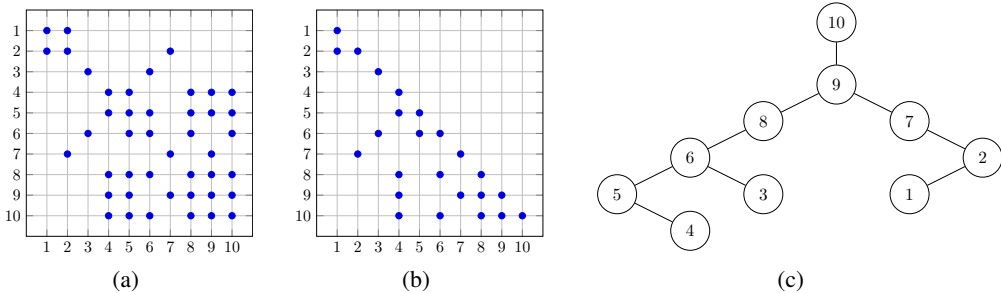

Figure 4: (a) A $10 \times 10$ positive definite matrix $P = AA^\top$, where the blue dot represent the non-zero elements in $P$. (b) The Cholesky factor $L$ of $AA^\top$. (c) The corresponding elimination tree for matrix $P$, where each node represent one column in the Cholesky factor. We can see that, as the row index of the first subdiagonal nonzero entry of the 6-th column is 8, the parent of node 6 is 8. Furthermore, the non-zero pattern of this coloumn is $\{6, 8, 10\}$, which is a subset of vertices on the path from node 6 to the root in the elimination tree.

$$\leq \frac{1}{T} \log(n/d)$$
$$\leq \frac{\epsilon}{50}$$
$$\leq \log(1 + \epsilon)$$

where the first step uses the definition of $\sigma_i(u)$, the second step follows from Corollary G.1, the third step follows from calculation, and the last step follows from the fact that for small $\epsilon$, $\epsilon/50 \leq \log(1+\epsilon)$. In conclusion, $\sigma_i(u) \leq 1 + \epsilon$.

Additionally, since for $k \in [T]$, each row of $w_{k,i}$ is a leverage score of some matrix, according to Lemma 2.2, we have:

$$\sum_{i=1}^{n} u_i = \sum_{i=1}^{n} \frac{1}{T} \sum_{k=1}^{T} w_{k,i}$$
$$= \frac{1}{T} \sum_{k=1}^{T} \sum_{i=1}^{n} w_{k,i}$$
$$= \frac{1}{T} \sum_{k=1}^{T} d$$
$$= \frac{1}{T} T d$$
$$= d$$

where the first line uses the definition of $u$, the second step follows from reorganization, the third step follows from Lemma 2.2, the fourth and the final step comes from calculation.

Thus, we complete the proof. $\qquad\square$

## G.2  Running Time

The rest of this section is to prove the running time of Algorithm 3. We first show the time needed to compute the leverage score with small treewidth setting.

**Lemma G.3.** *Given the Cholesky factorization $LL^\top$. Let $a_i^\top$ denote the $i$-th row of $A$, for each $i \in [n]$. Let $B = \sqrt{H} A \in \mathbb{R}^{n \times d}$ where $H$ is a nonnegative diagonal matrix. Let $\sigma_i = b_i^\top (B^\top B)^{-1} b_i$. We can compute $\sigma \in \mathbb{R}^n$ in $O(n\tau^2)$ time.*

*Proof.* Let $LL^\top = B^\top B$ be Cholesky factorization decomposition. Then, we have
$$b_i^\top (B^\top B)^{-1} b_i = b_i^\top L^{-\top} L^{-1} b_i$$

$$= (L^{-1}b_i)^\top (L^{-1}b_i).$$

Using the property of elimination tree, we have each row of $B$ has sparsity $\tau$ and they lie on a path of elimination tree $\mathcal{T}$. In this light, we are able to output $L^{-1}b_i$ in $O(\tau^2)$ time, and then compute a solution of sparsity $O(\tau)$.

Therefore, we can compute the score for a single column in $O(\tau^2)$. In total, it takes $O(n\tau^2)$. $\qquad\square$

Next, we show our main result.

**Theorem G.4** (Performance of Algorithm 3, formal version of Theorem 6.1). *For all $\epsilon \in (0, 1)$, we can find a $(1 + \epsilon)$-approximation of John Ellipsoid defined by matrix $A$ with treewidth $\tau$ inside a symmetric convex polytope in time $O((n\tau^2) \cdot T)$ where $T = \epsilon^{-1} \log(n/d)$.*

*Proof.* At first, initializing the vector $w$ takes $O(n)$ time. In the main loop, the per iteration running time can be decomposed as follows:

- Using Lemma 2.7, calculating the Cholesky decomposition for $B_k^\top B_k$ takes $O(n\tau^2)$ time.

- Using Lemma G.3, computing $w_{k+1}$ takes $O(n\tau^2)$ time.

Hence, the overall per iteration running time for the main loop is $O(n\tau^2)$ time, hence yields the total running time for the main loop as $O((n\tau^2)T)$.

Then, computing the average of vector $w$ from time $1$ to $T$, and computing the vector $v_i$ takes $O(nT)$ time. Finally, note that we don't have to output $A^\top V A$. Instead, we can just output $A$ and vector $v$, which takes $O(n)$ time.

Therefore, by calculation, the running time of Algorithm 3 is: $O((n\tau^2)T)$. Thus, we complete the proof. $\qquad\square$

## H   Limitations

While our findings primarily revolve around algorithmic advancements, we also see potential in exploring a matching lower bound for this problem in future research.

## I   Impact Statement

Our paper introduces research aimed at advancing the area of Machine Learning and Optimization. While there are numerous societal implications associated with our research, we believe none require particular emphasis in this context. We propose two algorithms that solve the John Ellipsoid problem more efficiently. We hope our work can inspire effective algorithm design and promote a better understanding of John Ellipsoid problem and the D-optimal design problem. Since this is a theoretical paper, we do not foresee any potential negative societal impact.

