# OpenReview forum: "Faster Algorithms for Structured John Ellipsoid Computation"
_NeurIPS.cc/2025/Conference — NeurIPS 2025 poster_

### Official Review · Reviewer_Aseo · 2025-06-01

**Clarity:** 3
**Significance:** 3
**Originality:** 3
**Rating:** 4
**Confidence:** 2

**Summary:**

This work proposes two new algorithms that accelerate the approximation of the John Ellipsoid within a symmetric polytope, claiming an improvement over the state-of-the-art CCLY19 method.
* The first algorithm utilizes the sparsity of the constraint matrix to lower the per-iteration complexity to $\tilde{\mathcal{O}}(\epsilon^{-1}\text{nnz}(A) + \epsilon^{-2}d^\omega)$, where $\omega$ is the matrix multiplication exponent.
* The second algorithm considers from the structural graph point of view, reduces this complexity further to $\mathcal{O}(n\tau^2)$ per iteration, with $\tau$ signifying the treewidth.

**Questions:**

- The improvement offered by Algorithm 1 over CCLY19 appears incremental, particularly since CCLY19 also discussed using sketching methods for acceleration. Could the authors elaborate on the primary novel contributions of Algorithm 1 compared to CCLY19?

- Could the authors provide concrete examples to illustrate the motivations for, and benefits of, using Algorithm 2?

**Ethical Concerns:**

["NO or VERY MINOR ethics concerns only"]

**Final Justification:**

The authors' response has effectively alleviated my concerns about the technical novelty of this work relative to [CCLY19]. While I am now confident in the paper's core contribution, the quality of the presentation still has room to improve. My recommendation is borderline accept.

**Limitations:**

NaN

**Paper Formatting Concerns:**

NaN

**Quality:**

3

**Strengths And Weaknesses:**

Strengths:
- This work proposes two interesting extensions based on previous work [CCLY19], and the theoretical discussion appears solid.
- The combination of John Ellipsoid estimation with the treewidth of a matrix-dual-graph seems new.

---
Weaknesses:
- Theorem 1.1 is somewhat confusing: the variable $\delta$ is introduced in the preceding text but is not explicitly part of the theorem's statement. Furthermore, it's unclear whether the theorem makes a probabilistic or deterministic claim.
- The motivation for incorporating treewidth could be further clarified. Real-world examples would help readers better understand its practical relevance.
- The primary audience for this work might align more with the TCS community than the NeurIPS audience, though this is a minor concern.

---

> ### Author Rebuttal · Authors · 2025-07-31
>
> Thanks for your thoughtful review. We appreciate your constructive feedback and the opportunity to clarify our contributions.
>
> ## Regarding Theorem 1.1 clarity
> We agree that we can further clarify this statement. The algorithm makes a probabilistic claim - it succeeds with probability at least $1-\delta-\delta_0$ where $\delta$ is the failure probability for leverage score approximation and $\delta_0$ is for the sampling process. We'll revise the theorem statement to explicitly include $\delta$ as a parameter and clearly state the probabilistic nature of the guarantee upfront. This will make it immediately clear that we're providing a randomized algorithm with high-probability bounds.
>
> ## Regarding motivation and applications and NeurIPS audience fit
>
> As mentioned in the introduction. John Ellipsoid is equivalent to the D-optimal design problem in statistics [1,2]. Moreover, John Ellipsoid has numerous applications in machine learning, including [3,4,5]. The faster solving algorithm of John Ellipsoid could also possibly accelerate these algorithms, thus improving the efficiency of real-world problems.
>
> ## Question 1: Novel contributions of Algorithm 1 vs [CCLY19]
>
> Thanks for asking us to clarify this important distinction. While CCLY19 does use sketching in their Algorithm 2, our contributions go significantly beyond their approach. CCLY19's sketching applies a Gaussian sketch $S \in \mathbb{R}^{s \times m}$ to reduce the per-iteration dependence on $m$ (number of constraints), but they still compute the full $d \times d$ matrix inverse $(B^T B)^{-1}$ exactly, resulting in $O(mn^2)$ per-iteration cost.
>
> Our key innovation is introducing leverage score sampling as an additional preprocessing step before sketching. This two-stage approach is novel: first, we compute approximate leverage scores and use them to subsample $O(\epsilon^{-2}d \log d)$ important rows from the $n \times d$ matrix, dramatically reducing the effective problem size; second, we apply sketching to this subsampled matrix, further accelerating computations.
>
> The combination enables us to work with much smaller matrices throughout, achieving $O(\epsilon^{-1}\text{nnz}(A) + \epsilon^{-2}d^{\omega})$ per iteration instead of CCLY19's $O(mn^2)$. For sparse matrices where $\text{nnz}(A) \ll mn$, this is a substantial improvement.
>
> Moreover, our novel telescoping analysis (Lemma 5.2) handles the interaction between sampling and sketching errors, which wasn't needed in CCLY19's simpler sketching-only approach. In essence, CCLY19 showed that sketching helps; we show that intelligent sampling before sketching helps much more, achieving the first input-sparsity time algorithm for this problem.
>
> ## Question 2: Concrete examples for Algorithm 2
>
> Beyond the real-world applications mentioned above, here's a concrete example: Consider solving a John ellipsoid problem for a transportation network with 10,000 nodes and tree-like structure (common in distribution networks). The constraint matrix would be $10,000 \times 10,000$ but with treewidth $\tau \approx 10$. Algorithm 2 would run in $O(10,000 \times 100) = 10^6$ operations per iteration, while CCLY19 needs $O(10,000^3) = 10^{12}$ operations - a million-fold improvement! We'll include such concrete examples to illustrate the practical impact.
>
> ## Conclusion
> We appreciate your engagement with our work and hope these clarifications address your concerns. We would like to kindly request if you could reconsider the rating for our paper, after our clarification. Thank you so much.
>
> ## Reference
> [1] Pukelsheim F. Optimal design of experiments. Society for Industrial and Applied Mathematics; 2006.
>
> [2] Todd MJ. Minimum-volume ellipsoids: Theory and algorithms. Society for Industrial and Applied Mathematics; 2016.
>
> [3] Allen-Zhu Z, Li Y, Singh A, Wang Y. Near-optimal design of experiments via regret minimization. ICML’17.
>
> [4] Wang Y, Yu AW, Singh A. On computationally tractable selection of experiments in measurement-constrained regression models. JMLR’17
>
> [5] Lu H, Freund RM, Nesterov Y. Relatively smooth convex optimization by first-order methods, and applications. SIAM Journal on Optimization, 2018.

---

> > ### Comment · Reviewer_Aseo · 2025-08-07
> >
> > I would like to thank the authors for their thorough response. The explanations provided, in conjunction with other reviewers' discussions, have successfully addressed my initial concerns regarding the work's contributions. I now agree that the methodology shows clear novelty beyond CCLY19, and I have updated my score to reflect this new assessment.

---

### Official Review · Reviewer_uqeL · 2025-06-25

**Clarity:** 2
**Significance:** 3
**Originality:** 3
**Rating:** 5
**Confidence:** 4

**Summary:**

The paper proposes faster algorithms for approximating the John ellipsoid, building upon and improving the best known result, by Cohen et al. (2019a). Namely, within the framework of Cohen et al., the authors propose two algorithms to improve the per-iteration cost: the first is based on improved sketching/sampling, and the second is based on treewidth.

**Questions:**

Presentation:
- Section 1
  - In both main theorems, please clarify the role of $\delta$ within the formal statement rather than after it for Theorem 1, and skipping it altogether for Theorem 2.
  - it would be nice to highlight the remark on L194 on the optimal upper/lower bounds somewhere in the introduction.
- Section 2
  - L150: treewidth of a ~tree~ graph is NP-hard
  - Definition 2.5
    - How's the block structure chosen?
    - Not sure I follow L155: submatrix of $A$ in column $i$
  - Lemma 2.7
    - It would help to clarify the case where $\tau$ is only known approximately
- Section 3
  - Eq(1): would you like to hint at how $\log(\det(G))^2$ captures the volume to be maximized?
  - Would it help to define $Q$ once, instead of twice on L188 and L192?
  - Is it possible to avoid the naming clash with the approximating polytope $Q$ in Lemma 3.4?
- Section 4
  - L212: would you like to highlight that the "accelerated algorithm" is based on **sketching**?
  - L216: this paragraph needs significant clarification
    - It starts off by digressing into a remark on rewriting $w$ as a leverage score.  While necessary, it delays the discussion of sampling, how sampling works, and why any of that benefits the fixed-point iteration.
   - Upon reaching "to be specific" on L220, the overview given is difficult to follow, and starts talking about a "sampling matrix"
   - The symbols $D$ and $K$, as well as $W$, are not defined.
   - Later on, it's seen that $D$ is the sampling matrix, but also $H$ and its approximate versions.
   - The aspect of overapproximating the leverage scores, the required rescaling, and the connection to the two references [CLM+15, DLS23] are not explained.
  - L234: this paragraph seems redundant, redefining notation.
    - Crucially though, it's where the vector $v$ appears, which ends up being the primary output of Algorithm 1.
  - L239: to further speed up the algorithm
    - What's the running time so far, before the additional speeding up?
  - L239: starts discussing Algorithm 1, out of no where, even pointing to specific lines.
  - Algorithm 1
    - Lines 6-9 can simply be replaced by $w_1 = d \cdot 1_n$
    - It would have helped to present a template pseudocode for the fixed-point iteration, leaving the iteration to a subprocedure, to allow easy comparison with the original algorithm of [CCLY19], and duplication in Algorithm 2.
    - On line#20, note that $\tilde{w}$ was not defined, and calling it original is pretty confusing.  Perhaps a reference to an equation or definition is better.
    - Note that line#32 is not useful.
    - Some line numbers are taken up just to accommodate comments.
- Section 5
  - As mentioned in the top-level comment, those informal statements end up taking space, defining and redefining various hyperparameters, while not contributing much towards comprehension and clarity cf. L265.
  - Section 5.2 is very difficult to appreciate, as the telescoping aspect was not highlighted in the algorithm to motivate this particular angle to the analysis.
  - L307: formal -> informal
  - L312: each row of $w_{k,i}$ is a leverage score of some matrix
    - The phrasing seems wrong.  Referring to line#13 in Algorithm 2, it should be easy to explicitly specify the matrix, instead of calling it some matrix.

Nitpicking:
- consider moving the placement of Table 1, e.g., at the top of the page
- L145: might as well define $E(G)$ explicitly.
- L256: during the iterative algorithm -> the fixed-point iteration in Eq(3)

Typos:
- L114: equivalently -> equivalent
- L125: propertie -> property
- L178: associate -> associated
- L206: for $a_i$ denote
- L218: we leverage the utilize
- L251: has some studied

**Ethical Concerns:**

["NO or VERY MINOR ethics concerns only"]

**Final Justification:**

The authors engaged in the rebuttal to convincingly address the presentation issues highlighted.  The technical content, motivation, and relevance were already satisfactory to me, so with the promise to revise the presentation I'm recommending acceptance.

**Limitations:**

Please mention the ease or feasibility of implementing those algorithms, to complement the theoretical progress and further substantiate the practical relevance of the approximate John ellipsoid.

**Quality:**

3

**Strengths And Weaknesses:**

Strengths:
- Improved algorithms for the approximation of the John ellipsoid for an important class of polytopes.  Besides being fundamental objects in math and algorithm theory, see also L18-23 for relevance to ML.
- The technical tools developed can find further applications, i.e., the approximation of leverage scores and matrix inverses; see L61-67.
- The connection to treewidth is an interesting perspective, and its practical manifestations can be appreciated more broadly; see L81-85.

Weaknesses:
- Although the earlier sections of the paper are well-written with good flow, the actual technical sections have serious issues.  Notably, the execution relies heavily on the appendices, resulting in a number of gaps in the overview Section 4, and reducing the informal statements for the analysis in Section 5 as hollow pointers to the appendices that otherwise don't contribute much.
- While I did not carefully review the appendix, it seems in much better shape, so the authors need to figure out how to summarize the main ideas in the paper.  To allow for this possibility, if the authors can convincingly recover in the rebuttal phase, I'm leaning to a borderline-accept score.
- If more space is needed, perhaps the second algorithm can be confined to the summary in the introduction, deferring the actual development to the appendix.

---

> ### Author Rebuttal · Authors · 2025-07-31
>
> Thanks for your exceptionally detailed and constructive review. We deeply appreciate your recognition of our technical contributions.
>
> ## Regarding reliance on appendices and Section 4 clarity
>
> ## Section 1 Questions
> (1) Role of $\delta$: We'll state this explicitly in the theorems rather than after.
>
> (2) Optimal bounds remark (L194): Thanks for the suggestion. We will revise it.
>
> ## Section 2 Questions:
> (1) L150: Thanks for pointing it out. We will add “graph” after “tree”.
>
> (2) Definition 2.5: The block structure partitions the $n$ rows into $m$ blocks. We'll clarify that this is given as part of the input when considering block-structured matrices.
>
> (3) L155: It is the submatrix of $A$ in column $i$ and row block $r$.
>
> (4) Lemma 2.7: We'll add that when only an $O(\tau \log^3 n)$-approximation is known (from the [BGS21] algorithm), our runtime becomes $O(n\tau^2 \log^6 n)$, which is still efficient for small $\tau$.
>
> ## Section 3 Questions
> (1) $\log(\det(G))^2$ and volume: We'll add that for an ellipsoid $\{x: x^TG^{-2}x \leq 1\}$, the volume is proportional to $\det(G^{-1})^{1/2} = \det(G)^{-1/2}$, so maximizing $\log(\det(G))^2 = 2\log(\det(G))$ maximizes volume.
>
> (2) Define Q once: Yes, we'll define Q once after equation (2) and reference it consistently.
>
> (3) Naming clash with Q: Good catch. We'll use $\mathcal{E}$ for the ellipsoid in Lemma 3.4 to avoid confusion.
>
> ## Section 4 Questions
> (1) L212 "accelerated algorithm": Yes, we'll explicitly state this refers to sketching-based acceleration.
>
> (2) L216 paragraph clarification: We'll completely rewrite this paragraph with clear flow: (i) explain that $w_{k+1,i}$ equals leverage scores of $B_k = \sqrt{W_k}A$, (ii) introduce leverage score sampling to approximate these scores efficiently, (iii) explain how oversampling by factor $\kappa$ ensures $(1\pm\epsilon_0)$ approximation.
>
> (3) Undefined symbols: We'll define all notation upfront: $D$ is the sampling matrix, $K$ is likely a typo (should be either $\mathcal{K}$ the constraint set or removed), $W = \text{diag}(w)$.
>
> (4) L234 redundancy: We'll remove the redundant definitions and introduce $v$ (the normalized output) more naturally.
>
> (5) L239 running time: We'll clarify that before sketching, the per-iteration cost is $O(\text{nnz}(A) + d^\omega)$ from leverage score computation.
>
> ## Algorithm 1 Issues
> We'll implement all your suggestions:
> - Replace lines 6-9 with $w_1 = (d/n) \cdot \mathbf{1}_n$
> - Add template pseudocode for comparison with CCLY19
> - Clarify that $\tilde{w}_{k+1,i}$ (line 20) approximates the ideal update $w_{k+1,i}$
> - Remove line 32 and consolidate comments
>
> ## Section 5 Issues
> (1) Section 5.2 telescoping: We'll add intuition explaining that telescoping tracks how approximation errors accumulate across iterations.
>
> (2) L307: Should be "informal" - we'll correct this.
>
> (3) L312 clarification: We'll specify explicitly that $w_{k,i}$ is the leverage score of the $i$-th row of matrix $B_k = \sqrt{W_k}A$.
>
> ## Typography and Typos
> Thanks for catching these - we'll fix all of them: L114, L125, L178, L206, L218, L251, and reposition Table 1.
>
> ## Limitations
> Regarding implementation feasibility: Our algorithms are practical to implement. The first algorithm uses standard randomized linear algebra primitives (Gaussian sketching, leverage score sampling) available in libraries like LAPACK/Scikit-learn. The treewidth algorithm leverages sparse Cholesky factorization, implemented efficiently in packages like CHOLMOD. We'll add this discussion to emphasize practical relevance.
>
> ## Conclusion
> We're grateful for your thorough review and your appreciation. Given that you found our technical contributions solid and indicated willingness to reconsider, we hope these planned improvements will earn your support for acceptance! Thank you so much.

---

> > ### Comment · Reviewer_uqeL · 2025-08-01
> > **Thanks for a constructive rebuttal**
> >
> > The suggested edits should account for the key requests highlighted.  As my score was mainly based on presentation issues, I'm raising the score to accept.  I hope the authors will follow through with the other reviewers regarding any concerns about the technical content.  Best of luck.

---

### Official Review · Reviewer_MGG9 · 2025-06-30

**Clarity:** 4
**Significance:** 3
**Originality:** 3
**Rating:** 4
**Confidence:** 1

**Summary:**

This paper considers the solution of the John Ellipsoid problem, which is to find the epplisoid of the largest volume that is contained inside a convex body. In particular it  assumes that the convex body is defined by a linear systems |A x| <= 1. It proposed two methods, one based on sketching that runs in near linear time to the input size, another tree-width based algorithm.

**Questions:**

- In Section 1.2, Table 1, if epsilon is small (<1/sqrt(n)), doesn’t it make the proposed algorithm slower than CCLY19 in cost per iter?

**Ethical Concerns:**

["NO or VERY MINOR ethics concerns only"]

**Limitations:**

yes

**Quality:**

3

**Strengths And Weaknesses:**

Strength:
- The paper is well organized, presenting the main results first, then hierarchically diving into the details
- The proof that leads to Theorem 1.1 is impressive and uses quite a bit of machinery
- The results do seem to be an improvement of previous results under certain conditions (e.g., how small is epsilon)

Weakness:
- For readers who are not familiar with this problem, it would be very helpful to explain the impact of the proposed algorithm in terms of implications to the applications in machine learning, e.g., would it be able solution of some ML problems that would otherwise be too slow?
- I don’t see a comparison with benchmarks in actual implementations. I believe Figure 2 shows theoretical results. How do the algorithms compare in practice?

---

> ### Author Rebuttal · Authors · 2025-07-31
>
> Thanks for your thorough review and constructive feedback. We truly appreciate your recognition of our work.
>
> ## Regarding empirical comparisons
>
> We focused on rigorous analysis of our algorithms' theoretical guarantees. We would like to clarify that our work is mostly a theoretical study, which does have a large place in the top machine learning conferences, like NeurIPS, ICLR, and ICML. For instance, NeurIPS accepts papers [1,2,4,5,6,7], all of which are entirely theoretical and do not contain any experiments. Moreover, ICLR and ICML also accepted papers [8,9], which are purely theoretical. Our work is similar to theoretical studies like [5,6,8,10], concentrating on the theoretical aspects of optimization algorithms.
>
> ## Regarding ML applications and impact
>
> As mentioned in Section 1, John Ellipsoid has numerous applications in machine learning, including [11,12,13]. The faster solving algorithm of John Ellipsoid could also possibly accelerate these algorithms, thus improving the efficiency of real-world problems.
>
> ## Regarding the epsilon parameter in Table 1
>
> Thank you for the question. You are correct. As we stated in Line 56 - 60,  when the matrix $A$ is dense, i.e.,$ \mathrm{nnz}(A) = \Theta(nd)$, our per-iteration cost becomes $O(\epsilon^{-1}nd +\epsilon^{−2}d^\omega)$. In the regime where $n > d^\omega$ and $\epsilon \in (1/d, 1)$, our algorithm is always better than [CCLY19] even when the matrix A is dense.
>
> ## Conclusion
>
> We're encouraged by your positive assessment. We hope these clarifications will solidify your confidence in accepting our work. Thank you so much.
>
> ## Reference
> [1] Alman, J., & Song, Z. Fast attention requires bounded entries. NeurIPS’23.
>
> [2] Alman J., & Song Z. The fine-grained complexity of gradient computation for training large language models. NeurIPS’24.
>
> [3] Alman, J., & Song, Z. How to capture higher-order correlations? generalizing matrix softmax attention to kronecker computation. ICLR’24.
>
> [4] Sarlos, T., Song, X., Woodruff, D., & Zhang, R. Hardness of low rank approximation of entrywise transformed matrix products. NeurIPS’23.
>
> [5] Dexter, G., Drineas, P., Woodruff, D., & Yasuda, T. Sketching algorithms for sparse dictionary learning: PTAS and turnstile streaming. NeurIPS’23.
>
> [6] Xu, Z., Song, Z. and Shrivastava, A., 2021. Breaking the linear iteration cost barrier for some well-known conditional gradient methods using maxip data-structures. NeurIPS’21.
>
> [7] Song, Z., Vakilian, A., Woodruff, D., & Zhou, S. On Socially Fair Low-Rank Approximation and Column Subset Selection. NeurIPS’24.
>
> [8] Song, Z., Yu, Z.: Oblivious Sketching-based Central Path Method for Linear Programming. ICML’21.
>
> [9] Brand JV, Song Z, Zhou T. Algorithm and hardness for dynamic attention maintenance in large language models. ICML’24.
>
> [10] Song Z, Yang X, Yang Y, Zhang L. Sketching meets differential privacy: fast algorithm for dynamic kronecker projection maintenance. ICML’23.
>
> [11] Allen-Zhu Z, Li Y, Singh A, Wang Y. Near-optimal design of experiments via regret minimization. ICML’17.
>
> [12] Wang Y, Yu AW, Singh A. On computationally tractable selection of experiments in measurement-constrained regression models. JMLR’17
>
> [13] Lu H, Freund RM, Nesterov Y. Relatively smooth convex optimization by first-order methods, and applications. SIAM Journal on Optimization, 2018.

---

> > ### Comment · Reviewer_mXV8 · 2025-08-05
> >
> > Thank you for your comments. I'm adjusting my score (provided that presentation issues will be fixed).

---

### Official Review · Reviewer_mXV8 · 2025-07-03

**Clarity:** 1
**Significance:** 3
**Originality:** 3
**Rating:** 4
**Confidence:** 3

**Summary:**

Computing the John ellipsoid of a convex set is a classical and computationally challenging problem in computer science. In certain cases where the convex bodies exhibit additional structure, more efficient algorithms are known. This paper focuses on the setting where the convex body is a convex and centrally symmetric polytope, and proposes efficient algorithms that refine the approach of CCLY’19. The key idea is to extend the simple fixed-point iteration scheme of CCLY’19 under either of two structural assumptions: (i) when the matrix defining the convex and centrally symmetric polytope is sparse, and (ii) when its dual graph has small treewidth.

**Questions:**

Could the authors provide concrete examples of practical problems in which the computation of the John ellipsoid is relevant and the defining matrix of the convex and centrally symmetric polytope is naturally sparse?

**Ethical Concerns:**

["NO or VERY MINOR ethics concerns only"]

**Final Justification:**

Improved score based on presentation improvements that the authors suggested

**Limitations:**

Limitations appropriately addressed. NA for societal impacts

**Quality:**

2

**Strengths And Weaknesses:**

Strengths and Weaknesses:

Overall, I find the presented results interesting and relevant within the literature on John ellipsoid computation. The required techniques are sufficiently novel, and, if I understand correctly, the assumption that the convex set is a centrally symmetric polytope is not overly restrictive and can have meaningful practical applications.


The quality of the results themselves appears adequate. The paper provides appropriate approximation guarantees and runtime analyses. I briefly reviewed some of the proofs, and they appear reasonable.


My primary concerns, however, relate to the quality of the presentation. The manuscript contains numerous confusing or unnatural passages, repeated definitions (for example, the paragraph beginning at line 234), as well as undefined terms (such as $n_i$​ in the definition of the dual graph). Additionally, there are many punctuation errors and typographical mistakes throughout the text, which may be artifacts of extensive rewriting or abbreviation. Furthermore, the main body of the text is not fully self-contained: for instance, the proof of Theorem 6.2 relies on Lemma G.2 from the appendix without sufficient explanation. Despite the technical contributions being of interest, these presentation issues create an overall impression of a manuscript that is unpolished and not yet ready for publication.


Another point I found confusing is that, while the authors frequently reference and build on CCLY’19, they do not provide a concise and coherent overview of the general structure of the CCLY approach itself. Personally, I found it difficult to keep track of how the new contributions relate to the existing framework, which made it challenging to separate what is directly adopted, what is modified, and what is entirely new. In my opinion, providing a clear and structured overview of the CCLY method early in the paper would greatly improve readability and help orient the reader more effectively.


At present, I am on the fence about recommending acceptance. Given the unclear and insufficiently polished presentation, I am inclined to suggest rejection, although I could be persuaded otherwise if the authors are able to significantly improve the clarity and self-contained nature of the paper.

---

> ### Author Rebuttal · Authors · 2025-07-31
>
> Thanks for your thorough review and constructive feedback. We truly appreciate your recognition of significance and quality.
>
> ## Regarding presentation quality
> We sincerely apologize for the numerous issues you encountered.
> We will conduct a comprehensive revision to address all presentation concerns, including fixing the repeated definitions, defining all terms clearly, and correcting all typos.
> We recognize that clear presentation is crucial for delivering our technical contributions effectively, and we're committed to revise the manuscript to publication-ready standards for the camera ready version.
>
> ## Regarding issues with [CCLY19]
> We agree with your suggestion about providing a clearer overview of the [CCLY19] framework. We will introduce [CCLY19] in a dedicated section in our revised version that comprehensively explains its structure, adopted components, modified components and our new contribution.
>
> ## Issues about Theorem 6.2
> Regarding the self-containedness issue with Theorem 6.2, we will ensure that all key lemmas referenced from the appendix are either briefly stated in the main text or their intuition is sufficiently explained.
>
> ## Regarding practical applications
>
> As mentioned in the introduction. John Ellipsoid is equivalent to the D-optimal design problem in statistics [1,2]. Moreover, John Ellipsoid has numerous applications in machine learning, including [3,4,5]. The faster solving algorithm of John Ellipsoid could also possibly accelerate these algorithms, thus improving the efficiency of real-world problems.
>
> ## Conclusion
> We truly appreciate your elaborate and detailed feedback. Given that you find our technical contributions of interest with appropriate guarantees and novel techniques, we hope you'll reconsider your rating after we address all presentation issues! Thank you so much.
>
> ## Reference
>
> [1] Pukelsheim F. Optimal design of experiments. Society for Industrial and Applied Mathematics; 2006.
>
> [2] Todd MJ. Minimum-volume ellipsoids: Theory and algorithms. Society for Industrial and Applied Mathematics; 2016.
>
> [3] Allen-Zhu Z, Li Y, Singh A, Wang Y. Near-optimal design of experiments via regret minimization. ICML’17.
>
> [4] Wang Y, Yu AW, Singh A. On computationally tractable selection of experiments in measurement-constrained regression models. JMLR’17
>
> [5] Lu H, Freund RM, Nesterov Y. Relatively smooth convex optimization by first-order methods, and applications. SIAM Journal on Optimization, 2018.

---

### Note · Authors · 2025-08-15

We thank the reviewers for their thoughtful assessments. Reviewers recognized that our work advances faster algorithms for structured John-ellipsoid computation, with sufficiently novel techniques, appropriate approximation guarantees, and meaningful practical relevance; they also noted the new perspective via treewidth. One reviewer further highlighted the paper’s organization and the impressive proof of Theorem 1.1.

All questions raised in the reviews have been addressed in our rebuttal. We clarified the $\epsilon$–related regimes and Table 1 tradeoffs, and we explained the scope and impact in ML terms, emphasizing that this is a theoretical contribution in line with prior accepted theory papers. We also described how our results relate to and build upon the CCLY’19 framework. For the detailed presentation questions, we provided point-by-point clarifications across Sections 1–5 and Algorithm 1 (e.g., explicitly stating theorem parameters, reorganizing Section 4’s flow and notation, and explaining the telescoping argument). Finally, we added a concrete discussion of implementation feasibility in the Limitations section.

Because the rebuttal format does not allow uploading a revised manuscript, we did not submit an updated version during the discussion period. In the revision, we will implement exactly what we stated in the rebuttal: a comprehensive presentation pass to fix repeated definitions, define all terms, and correct typos; a clearer overview of CCLY’19; making Theorem 6.2 self-contained; rewriting Section 4 for readability with all notation defined up front; restructuring Algorithm 1 as suggested; adding intuition for the telescoping argument; repositioning Table 1; and including the implementation-feasibility note in Limitations.

We appreciate your comments and the constructive guidance.

---

### Decision · Program_Chairs · 2025-09-17

**Decision:**

Accept (poster)

**Comment:**

The authors present a well-motivated problem, provide a novel method to address it, and provide a nontrivial proof to guarantee their method’s performance. The tools they introduce have the potential to be used more broadly. The authors must clean up their writing and polish their presentation. They also need to provide a clearer overview of [CCLY19].